# Annotation Efficiency: Identifying Hard Samples via Blocked Sparse Linear Bandits

## Abstract

This paper considers the problem of annotating datapoints using an expert with only a few annotation rounds in a *label-scarce* setting. We propose soliciting reliable feedback on difficulty in annotating a datapoint from the expert in addition to ground truth label. Existing literature in active learning or coreset selection turns out to be less relevant to our setting since they presume the existence of a reliable trained model, which is absent in the label-scarce regime. However, the literature on coreset selection emphasizes the presence of difficult data points in the training set to perform supervised learning in downstream tasks (Mindermann et al., 2022). Therefore, for a given fixed annotation budget of $\mathsf{T}$ rounds, we model the sequential decision-making problem of which (difficult) datapoints to choose for annotation in a sparse linear bandits framework with the constraint that no arm can be pulled more than once (*blocking constraint*). With mild assumptions on the datapoints, our (computationally efficient) Explore-Then-Commit algorithm $\texttt{BSLB}$ achieves a regret guarantee of $\widetilde{\mathsf{O}}(k^{\frac{1}{3}}\mathsf{T}^{\frac{2}{3}} + k^{-\frac{1}{2}}\beta_k + k^{-\frac{1}{12}}\beta_k^{\frac{1}{2}}\mathsf{T}^{\frac{5}{6}})$ where the unknown parameter vector has tail magnitude $\beta_k$ at sparsity level $k$. To this end, we show offline statistical guarantees of Lasso estimator with mild Restricted Eigenvalue (RE) condition that is also robust to sparsity. Finally, we propose a meta-algorithm $\texttt{C-BSLB}$ that does not need knowledge of the optimal sparsity parameters at a no-regret cost. We demonstrate the efficacy of our $\texttt{BSLB}$ algorithm for annotation in the label-scarce setting for an image classification task on the PASCAL-VOC dataset, where we use real-world annotation difficulty scores.

## 1 Introduction

In niche industrial applications such as Named Entity Recognition (Nguyen et al., 2023) and learning tasks on low-resource languages (Hedderich et al., 2021), obtaining high-quality labels is challenging due to the lack of expert annotators. However, high-quality labels are critical for effective model training (Li et al., 2023), and thus, expert annotators must provide ground truth labels. (Sorscher et al., 2022) demonstrated that selecting high-quality data can reduce the power-law association of test error with dataset size to an exponential law. In annotation-expensive tasks with large volumes of unlabeled data, the challenge is to select a representative subset of datapoints for labeling. In label-scarce tasks, where the number of expert annotators is extremely low, often only one, it is impractical to query the same datapoint multiple times. While crowd-sourcing literature reduces noise by aggregating labels from multiple annotators (Verroios & Garcia-Molina, 2015), the annotation from a single or aggregated expert is considered the final ground truth label in our setting. We term this restriction, where a datapoint cannot be re-queried after annotation, the *blocking constraint*. Additionally, the annotation budget is typically much smaller than the datapoint embedding dimension. An efficient annotation strategy should be sequential (instead of one-shot) in such cases, as each annotation informs future decisions and helps identify more informative datapoints.

In addition, data-pruning techniques like coreset selection emphasize selecting hard examples (Maharana et al., 2023), while (Sorscher et al., 2022) justifies this for perceptron learning. Curriculum learning (Bengio et al., 2009) also uses increasingly difficult examples, but defining the 'hardness' of unlabeled data is ambiguous. Hard examples identified by heuristics are often noisy, mislabeled, or outliers (Mindermann et al., 2022). *To address this, we propose soliciting annotation difficulty feedback directly from expert annotators*. This adds minimal annotation overhead and helps identify

hard examples more reliably than heuristics. Additionally, leveraging expert feedback to learn and predict hardness (via an auxillary model) is simpler than addressing complex downstream task itself.

Thus, the primary problem we aim to solve is: *"Given a small annotation budget* $\mathsf{T}$*, how should we sequentially identify unique and difficult unlabeled datapoints for annotation as we receive expert feedback on hardness of annotation?"* Theoretically, we model this sequential decision-making process using *sparse linear bandits with a blocking constraint*. Specifically, suppose we have a large set $\mathcal{A}$ of M unlabeled datapoints, each associated with a $d$-dimensional embedding. With $\mathsf{T}$ rounds available, at each round $t \in [\mathsf{T}]$, a datapoint $\mathbf{a}_t$ is selected for annotation, and the expert provides noisy feedback $r_t$ on the difficulty of labeling that datapoint, along with the ground truth label. To address the limited annotation budget and the relevance of a few important features in high-dimensional spaces (Hao et al., 2020), we model the expected hardness as a sparse linear function of the embedding. We ensure robustness to sparsity - our theoretical guarantees degrade gracefully as the parameter vector becomes less sparse, i.e., as it deviates from sparsity due to an increasing tail.

Although our primary motivation comes from data selection for annotation in a label-scarce regime, our theoretical framework is broadly applicable. For example, resource-constrained edge devices, such as smartphones, consider the problem of recommendation of personalized products. In applications such as movie/book recommendations, typical users will hardly consume an item more than once - so any feedback provided is not subject to change. Furthermore, the user will provide feedback for every item (books/movies) in a large repository of items. Here, our theoretical framework (sparse linear bandits with *blocking constraint*) is directly applicable as it involves learning unknown user taste from sequential item recommendations based on user feedback. While existing literature addresses this by combining feedback from multiple users (Bresler et al., 2014; Pal et al., 2024), such algorithms compromise user privacy. Our framework will not have this limitation.

**Notation:** We denote vectors by bold small letters (say $\mathbf{x}$), scalars by plain letters (say $x$ or $\mathsf{X}$), sets by curly capital letters (say $\mathcal{X}$) and matrices by bold capital letters (say $\mathbf{X}$). We use $[m]$ to denote the set $\{1, 2, \ldots, m\}$, $\|\mathbf{x}\|_p$ to denote the $p$-norm of vector $\mathbf{x}$. For a set $\mathcal{T}$ of indices $\mathbf{v}_{\mathcal{T}}$ is used to denote the sub-vector of $\mathbf{v}$ restricted to the indices in $\mathcal{T}$. $\lambda_{\min}(\mathbf{A})$ denotes the minimum eigenvalue of the matrix $\mathbf{A}$ and $\mathsf{diag}(\mathbf{x})$ denotes a diagonal matrix with entries as $\mathbf{x}$. We use $\mathcal{B}^d$ and $\mathcal{S}^{d-1}$ to denote the unit ball and unit sphere in $d$ dimensions, respectively. We will write $\mathbb{E}X$ to denote the expectation of a random variable $X$. $\widetilde{O}(\cdot)$ notation hides logarithmic factors in $\mathsf{T}, \mathsf{M}, d$.

## 1.1 PROBLEM FORMULATION AND PRELIMINARIES

Consider a dataset with a set (or more generally, a multi-set) of M unlabeled data-points $\mathcal{A} \equiv \{\mathbf{a}^{(1)}, \mathbf{a}^{(2)}, \ldots, \mathbf{a}^{(\mathsf{M})}\} \subseteq \mathbb{R}^d$, each of which will be referred to as an *arm* in the bandit setup. Let $\mathbf{a}^{(j)}$ denote the $d$-dimensional vector embedding associated with the $j^{\mathsf{th}}$ data-point (arm). The arm embedding vectors are contained in a ball of radius R that is $\|\mathbf{a}\|_2 \leq \mathsf{R} \,\forall \mathbf{a} \in \mathcal{A}$.

We have an annotation budget of $\mathsf{T}$ rounds. In our annotation-poor regime, we have $\mathsf{T} \ll d \ll \mathsf{M}$; that is, the annotation budget is much smaller than the ambient dimension, which, in turn, is significantly smaller than the number of arms. At each round $t \in [\mathsf{T}]$, an unlabelled data-point $\mathbf{a}_t$ (corresponding to the arm which has not been pulled in the first $t-1$ rounds) is selected by the online algorithm (decision-maker) and sent to the expert annotator for labeling. Note that such a selection mechanism respects the blocking constraint, which stipulates that no unlabeled datapoint will be sent for annotation more than once. The expert [1], as feedback, labels the datapoint $\mathbf{a}_t$ and provides $r_t$ that corresponds to the difficulty experienced in providing the ground truth label for $\mathbf{a}_t$. We model the expected hardness $\mathbb{E}r_t\langle\boldsymbol{\theta}, \mathbf{a}_t\rangle$ as a linear function of the arm embedding where $\boldsymbol{\theta} \in \mathbb{R}^d$ is an unknown parameter vector. In particular, the random variable $r_t$ is obtained as $r_t = \boldsymbol{\theta}^{\mathsf{T}}\mathbf{a}_t + \eta_t$ where $\eta_t$ is zero-mean i.i.d noise random variable with bounded variance $\sigma^2$. More precisely,

1. $\mathbb{E}[\eta_t | \mathcal{F}_t] = 0$ and $\mathbb{E}[\eta_t^2 \mid \mathcal{F}_t] \leq \sigma^2$, where $\mathcal{F}_t = \{(\mathbf{a}_1, r_1), \ldots (\mathbf{a}_{(t-1)}, r_{(t-1)})\}$ denotes the filtration till round $t \in [\mathsf{T}]$.
2. For any sparsity level $k \leq d$ we define the tail of the parameter vector $\boldsymbol{\theta}$ as, $\beta_k := \|\boldsymbol{\theta}_{\mathcal{T}_k^c}\|_1$ where $\mathcal{T}_k$ denotes the set of $k$ largest coordinates of $\boldsymbol{\theta}$ by absolute value and $\mathcal{T}_k^c = [d] \setminus \mathcal{T}_k$.

---

[1]If there are multiple experts, we can consider the final feedback to be aggregated, which does not change.

Note that in the special case of $k = 0$ for some sparsity level $k \ll d$, $\boldsymbol{\theta}$ will be referred to as a hard-sparse vector. In our set-up, we account for soft sparsity when the tail is non-zero and unknown - our statistical guarantees degrade gracefully as the tail magnitude increases. Now, we formally define the objective (commonly known as regret) in our online learning set-up, which also respects the *blocking constraint*. Our regret definition captures the difference in cumulative expected hardness of datapoints selected by the online algorithm versus the cumulative expected hardness of top $\mathsf{T}$ unique datapoints. Consider a permutation $\pi : [|\mathcal{A}|] \to [|\mathcal{A}|]$ of arms such that for any $i < j$, we have $\langle \boldsymbol{\theta}, \mathbf{a}^{(\pi(i))} \rangle \geq \langle \boldsymbol{\theta}, \mathbf{a}^{\pi(j)} \rangle$. We define the regret $\mathsf{Reg}(\mathsf{T})$ for our setting as,

$$\mathsf{Reg}(\mathsf{T}) := \sum_{t=1}^{\mathsf{T}} \langle \boldsymbol{\theta}, \mathbf{a}^{(\pi(t))} \rangle - \sum_{t=1}^{\mathsf{T}} \langle \boldsymbol{\theta}, \mathbf{a}_t \rangle. \tag{1}$$

We aim to design an algorithm that minimizes expected regret $\mathbb{E}[\mathsf{Reg}(\mathsf{T})]$ where the expectation is over the randomness in the algorithm.

**Our Contributions and Techniques:** The sparse linear bandits framework in the data-sparse regime was first studied by Hao et al. (2020); Jang et al. (2022) - however, these works do not consider either the blocking constraint or robustness to sparsity - both of which pose novel technical challenges. We propose BSLB (Blocked Sparse Linear Bandits), an efficient algorithm in this setting which is primarily an Explore-Then-Commit algorithm. In the exploration period, we carefully choose a subset of arms from which we sample without replacement - the goal is to ensure that the expected covariance matrix has a large minimum eigenvalue. In the exploitation period, we use a lasso estimator to estimate the unknown model parameters while being robust to sparsity. The optimal exploration period depends on the correct sparsity level of the unknown parameter vector, which is difficult to set in practice. Therefore, we also present a meta-bandit algorithm based on corralling a set of base bandits algorithms (Agarwal et al., 2017) - obtaining the same order-wise regret guarantees but without needing the knowledge of hyperparameters. Below, we summarize our *main contributions*:

1. We theoretically model the problem of identifying hard (informative) unlabeled datapoints by high dimensional sparse linear bandits framework with blocking constraint - the number of rounds or annotation budget is very small, even smaller than the ambient dimension. The hardness is modeled as a sparse linear function of the datapoint embedding - we allow for robustness to sparsity in the parameters where the tail magnitude is non-zero and unknown.

2. We propose an efficient "explore then commit" (ETC) algorithm BSLB for regret minimization in our framework (Theorem 2) that achieves a regret guarantee of $O(k^{1/3}\mathsf{T}^{2/3} + k^{-1/12}\beta_k^{1/2}\mathsf{T}^{5/6})$ for a fixed sparsity $k$ and the corresponding tail magnitude $\beta_k$ - for the special case of hard sparsity (tail magnitude $\beta_k = 0$), BSLB achieves a regret guarantee of $O(k^{1/3}\mathsf{T}^{2/3})$ (Corollary 2). The run-time of BSLB is polynomial in the number of datapoints and is therefore efficient.

3. A novel ingredient (function GETGOODSUBSET()) in the proposed algorithm BSLB is to select a *good* set of representative datapoints from the large set of unlabelled datapoints. This objective entails solving a discrete non-convex optimization problem in Equation 2. We propose a convex relaxation of the objective and obtain a feasible solution via randomized rounding - Theorem 3 states that our computationally efficient algorithm has good approximation guarantees.

4. BSLB requires knowledge of sparsity $k$ (which also controls the tail magnitude $\beta_k$) to set exploration period length. Since this is challenging to set in practice, we propose a meta-algorithm C-BSLB that combines base algorithms with different exploration periods. C-BSLB achieves same regret guarantees order-wise (Theorem 4) as BSTB but without knowledge of sparsity.

5. To validate our hypothesis and theory, we present experiments on the public image classification dataset PASCAL VOC 2012. Additional experiments on text classification on SST-2 and personalized recommendation with MovieLens, Jester, and Goodbooks datasets are in Appendix A.2.

**Technical Challenges:** At a high level, our proof follows a similar structure as in the analysis of Algorithm ESTC proposed in (Hao et al., 2020). First of all, both in (Hao et al., 2020) and our setting, there are no assumptions on the datapoint embeddings - unlike most existing literature on sparsity where the Gram matrix is assumed to satisfy Restricted Isometry Property (RIP) (Boche et al., 2015). A much weaker condition (namely Restricted Eigenvalue (RE)) in characterizing nice statistical guarantees of the Lasso estimator for sparse linear regression was shown in (Bickel et al., 2009; Rudelson & Zhou, 2013) - it was shown that if the covariance matrix of a distribution satisfies RE, then a sufficient number of independent samples will also satisfy RE. However, in our setting, the additional blocking constraint and our desired robustness to sparsity present three significant

technical challenges. (A) **Sampling:** While the above techniques can be used directly in the analysis of Algorithm ESTC, the blocking constraint in our setting entails that sampling of datapoints cannot be independent in the exploration phase. (B) **Statistical guarantees:** To the best of our knowledge, when the Gram matrix only satisfies RE, the statistical guarantees of the Lasso estimator hold only for hard sparse vectors - for these vectors, all but $k \ll d$ entries are zero. Existing guarantees for soft sparsity (non-zero tail) hold only for Gram matrices satisfying the stronger RIP condition (Wainwright, 2019). (C) **Knowledge of hyper-parameters:** Our first proposed algorithm BSLB (similar to ESTC) requires as input a fixed sparsity $k$ (corresponding tail magnitude $\beta_k$). If the input sparsity is too low or too high, then the regret guarantee will not be reasonable. However, it is challenging to set the sparsity parameter without having the knowledge of parameter vector itself. We resolve all technical challenges with novel techniques - in particular, the latter two might be of independent interest in other applications.

To resolve (A), we need to optimize for a probability distribution on datapoints (for sampling) with a well-conditioned expected covariance matrix under the constraint that it is uniform on a subset of datapoints and has zero mass on others. The related optimization problem is discrete and non-convex - we describe a convex relaxation of the objective and show a randomized rounding procedure to obtain a feasible solution with good approximation guarantees on the minimum eigenvalue of the expected covariance matrix. Subsequently, we show that subsampling from the recovered distribution in the exploration component ensures that the data covariance matrix satisfies RE with high probability. To resolve (B), we follow two steps 1) We extend guarantees in (Rudelson & Zhou, 2013) (Theorem 23) that demonstrates a variant of restricted isometry - the idea is that the data matrix acts as a near isometry on the image of all sparse vectors under a linear transformation. Such a technique, in turn, allows us to extend RE guarantees for design matrices whose rows are sampled without replacement 2) using the RE condition we derive guarantees for soft sparsity from the results on hard sparse vectors using an iterative approach to ensure robustness ((Boche et al., 2015) Theorem 1.6). For (C), we use a corralling algorithm based on the techniques of (Agarwal et al., 2017) that combines several base algorithms and provides guarantees with respect to the optimal one. However, a naive application leads to an additional corralling cost with a linear dependence on dimension $d$, making the regret vacuous ($\mathsf{T} \ll d$). We use a covering argument - we only choose a small representative subset (covering set) of base algorithms for input to corralling and show that for all remaining base algorithms, their regret is close enough to a base algorithm in the covering set.

**Related Work:** Conceptually our work is similar to *active learning* (Settles, 2009; Lesci & Vlachos, 2024) where unlabeled samples are annotated adaptively, based on the confidence of a trained model (Coleman et al., 2022). Active learning works well with good initialization and informative confidence intervals. However, in our label-scarce setting, active learning is particularly challenging with complex data due to the absence of a reliably trained model in the first place - this is more pronounced for difficult datapoints for which prediction is hard. Active Learning (AL) needs an initial set of high-quality labeled samples to reasonably train a model - also known as the *cold-start* problem - when labels are scarce, uncertainty based sampling techniques (AL) are unsuitable (Li et al., 2024). Our goal is to identify informative samples with the help of the expert annotator(s), whom the final model aims to emulate. *Coreset selection* (Guo et al., 2022; Albalak et al., 2024; Sener & Savarese, 2018) aims to select a subset of datapoints for training. However, coreset selection assumes that a large amount of *labeled data* already exists, and the focus is on reducing computational costs. In contrast, our setting deals with the lack of labeled data, making existing coreset selection approaches, which rely on the entire labeled dataset, inapplicable. Our work also aligns with *curriculum learning* (Bengio et al., 2009), where a model is trained on samples of increasing difficulty/complexity. Due to the ambiguity in hardness definition, often heuristics are used to infer the difficulty of samples (Soviany et al., 2022) and can turn out unreliable and not generalizable. *For niche tasks where an expert annotator is available, the difficulty ratings from the annotator are more informative since the goal is to train a model to mimic the expert.* In computer vision, there has been recent work regarding estimating the difficulty of a dataset for the model using implicit difficulty ratings of annotation (Ionescu et al., 2016; Mayo et al., 2023). For NLP tasks, Ethayarajh et al. (2022) constructs information theoretic metrics to estimate the difficulty of data points.

---

**Algorithm 1** Blocked Sparse Linear Bandits (`BSLB`) for Efficient Annotation

---

1: **Input: Unlabeled datapoints** $\mathcal{A}$, **Annotation Budget** $\mathsf{T}$, **Exploration Budget** $\mathsf{T}_{\text{explore}}$, **Regularization Parameter** $\lambda$, **Subset selection parameter** $\hat{g}$
2: $\mathcal{G} = \text{GETGOODSUBSET}(\mathcal{A}, \hat{g})$         ▷ Compute good subset of datapoints (arms)
3: $\mathcal{C} = \{\}, \mathcal{R} = \{\}$         ▷ Initialize Arm and Reward Set
4: **for** $t \in [\mathsf{T}_{\text{explore}}]$ **do**
5:     Sample randomly $\mathbf{a}_t \sim \mathcal{G}$ and get difficulty score $r_t$         ▷ Pull arm and get feedback
6:     $\mathcal{C} \leftarrow \mathcal{C} \cup \{\mathbf{a}_t\}, \ \ \mathcal{R} \leftarrow \mathcal{R} \cup \{r_t\}$         ▷ Store datapoint/arm and Difficulty Score
7:     $\mathcal{G} = \mathcal{G} \setminus \{\mathbf{a}_t\}$         ▷ Update unlabeled good subset
8: **end for**
9: $\widehat{\boldsymbol{\theta}} = \arg\min_{\boldsymbol{\theta}} ||\boldsymbol{\theta}||_1$ s.t. $\sum_{t \in [\mathsf{T}_{\text{explore}}]} (\mathcal{R}[t] - \langle \boldsymbol{\theta}, \mathcal{C}[t] \rangle)^2 \le \lambda$         ▷ Compute estimate using LASSO
10: $\mathcal{D} = \mathcal{A} \setminus \mathcal{C}$         ▷ Datapoints (arms) available for exploit phase
11: **for** $t \in [\mathsf{T}_{\text{explore}} + 1, \mathsf{T}]$ **do**         ▷ Take Top-$(\mathsf{T} - \mathsf{T}_{\text{explore}})$ difficult samples
12:     $\mathbf{a}_t = \arg\max_{\mathbf{a} \in \mathcal{D}} \widehat{\boldsymbol{\theta}}^\mathsf{T} \mathbf{a}$
13:     $\mathcal{C} \leftarrow \mathcal{C} \cup \{\mathbf{a}_t\}, \ \mathcal{D} = \mathcal{D} \setminus \{\mathbf{a}_t\}$
14: **end for**
15: **procedure** GETGOODSUBSET(Set of Samples $\mathcal{A}$, Subset selection parameter $\hat{g}$)
16:     **Output:** Sampled Subset $\mathcal{G}$
17:     Maximize the objective function defined in (6) with input $\hat{g}$ to obtain distribution $\hat{\boldsymbol{\mu}}$ over $\mathcal{A}$.
18:     **for** $j \in [\mathsf{M}]$ **do**
19:        $\mathcal{G} = \mathcal{G} \cup \{\mathbf{a}^{(j)}\}$ with probability $\hat{g}\hat{\boldsymbol{\mu}}_j$         ▷ Add sample $j$ to $\mathcal{G}$ with prob. $\hat{g}\hat{\boldsymbol{\mu}}_j$
20:     **end for**
21: **end procedure**

---

## 2    OUR ALGORITHM AND MAIN RESULTS

Our main contribution is to propose an Explore-Then-Commit (ETC) algorithm named `BSLB` which is summarized in Algorithm 1. `BSLB` takes as input a set of unlabeled datapoints (arms) $\mathcal{A}$, the annotation budget (time horizon) $\mathsf{T}$, the exploration budget $\mathsf{T}_{\text{explore}}$ and Subset selection parameter $\hat{g}$. Steps 2-8 of `BSLB` correspond to the exploration component in the algorithm. In Step 2, we first compute a good subset of arms $\mathcal{G} \subset \mathcal{A}$ (using function GETGOODSUBSET$(\mathcal{A}, \hat{g})$) which comprises of representative arms that cover the $d$-dimensional space reasonably well. Subsequently, in Steps 4-8, we sample arms without replacement from the set of arms $\mathcal{G}$ for $\mathsf{T}_{\text{explore}}$ rounds. The goal in the exploration component is to select a subset of arms such that the image of sparse vectors under the linear transformation by the gram matrix of the selected set has a sufficiently large magnitude (see Definition 1). As we prove, such a result ensures nice statistical guarantees of the parameter (difficulty) estimation with the subset labeled (with difficulty scores) at the end of the exploration component. Since the set of arms, $\mathcal{A}$ can be arbitrary, note that sampling arms uniformly at random from the entire set might not have good coverage - especially when most arms are concentrated in a lower-dimensional subspace. Therefore, finding a good representative subset of arms leads to the following discrete optimization problem

$$\lambda^*_{\min} := \max_{\mathcal{G}' \subseteq \mathcal{A}} \lambda_{\min} \left( |\mathcal{G}'|^{-1} \sum_{\mathbf{a} \in \mathcal{G}'} \mathbf{a}\mathbf{a}^\mathsf{T} \right). \tag{2}$$

The function GETGOODSUBSET$(\mathcal{A}, \hat{g})$ approximates the solution to this computationally infeasible discrete optimization. We maximize a relaxed concave program in equation 6 efficiently for a chosen input parameter $\hat{g}$ to obtain a distribution $\hat{\boldsymbol{\mu}}$ on the set of arms $\mathcal{A}$ - subsequently, we construct the subset $\mathcal{G}$ using randomized rounding (Step 19) with $\hat{\boldsymbol{\mu}}$ to obtain a feasible solution to 2.

The second part of `BSLB` (Steps 9-14) corresponds to the exploitation component of the algorithm. In Step 9, we use the Lasso estimator to get an estimate $\widehat{\boldsymbol{\theta}}$ of the unknown parameter vector $\boldsymbol{\theta} \in \mathbb{R}^d$. Note that the number of samples used in obtaining the estimate $\widehat{\boldsymbol{\theta}}$ is much smaller than the ambient dimension $d$. Finally, in Steps 11-14, we choose datapoints that are predicted to be hard according to our recovered estimate $\widehat{\boldsymbol{\theta}}$ and submit them for annotation. At every round in `BSLB`, no arm is pulled more than once, thus respecting the *blocking constraint*. It is important to note that `BSLB` is two-shot. That is, we change our data acquisition strategy only once after the exploration component - thus making our algorithm easy to use in practice. Next, we move on to our main theoretical results.

## 2.1 OFFLINE LASSO ESTIMATOR GUARANTEES WITH SOFT SPARSITY AND RE CONDITION

To the best of our knowledge, there do not exist in the literature offline guarantees for sparse linear regression that is (A) robust to sparsity modeling assumption and (B) holds only under the mild RE condition on the Gram matrix. Our first theoretical result fills this gap to a certain extent with an upper bound on error rate. We will start by introducing the definition of Restricted Eigenvalue (RE)

**Definition 1.** *Restricted Eigenvalue (RE):* $\mathbf{X} \in \mathbb{R}^{n \times d}$ *satisfies Restricted Eigenvalue property* $\mathsf{RE}(k_0, \gamma, \mathbf{X})$, *if there exists a constant* $K(k_0, \gamma, \mathbf{X})$ *such that for all* $z \in \mathbb{R}^d$ *and* $z \neq \mathbf{0}$,

$$0 < K(k_0, \gamma, \mathbf{X}) = \min_{J \subseteq \{1,...,d\} | |J| \leq k_0} \min_{\|z_{J^c}\|_1 \leq \gamma \|z_J\|_1} \frac{\|\mathbf{X}z\|_2}{\|z_J\|_2}.$$

Bickel et al. (2009) showed that RE is among the weakest conditions imposed in the literature on the Gram matrix to ensure nice statistical guarantees on the Lasso estimator for sparse linear regression.

**Theorem 1.** *Let* $\mathbf{X} \in \mathbb{R}^{n \times d}$ *be the data matrix with* $n$ *samples, dimension* $d$. *Let* $\mathbf{r} \in \mathbb{R}^n$ *be the corresponding observations such that* $\mathbf{r} = \mathbf{X}\boldsymbol{\theta} + \boldsymbol{\eta}$, *where* $\boldsymbol{\eta} \in \mathbb{R}^n$ *is a zero-mean random vector with i.i.d. components having bounded variance* $\sigma^2 = O(1)$. *Suppose* $\mathbf{X}$ *satisfies restricted eigenvalue property (Def. 1) with* $\mathsf{RE}(k, 4(1 + \gamma_1), \frac{\mathbf{X}}{\sqrt{n}})$ *with constant* $K$. *Let* $\boldsymbol{\theta}$ *have a tail* $\beta_k$ *at sparsity level* $k$ *that is,* $\beta_k := \|\boldsymbol{\theta}_{\mathcal{T}_k^c}\|_1 \leq \gamma_2 \|\boldsymbol{\theta}_{\mathcal{T}_k}\|_1$ *for some* $\gamma_2 \in \mathbb{R}$ *satisfying* $\gamma_2 \leq \gamma_1$, *where* $\mathcal{T}_k$ *is the set of* $k$ *largest coordinates by absolute value. An estimate* $\widehat{\boldsymbol{\theta}}$ *of* $\boldsymbol{\theta}$ *recovered using Lasso (Line 9 in* BSLB*), satisfies following with probability* $1 - \exp(-\Omega(n))$

$$\|\boldsymbol{\theta} - \widehat{\boldsymbol{\theta}}\|_2 = \widetilde{O}\left(n^{-1/2} k^{1/2} K^{-2} + k^{-1/2} \beta_k + n^{-1/4} \beta_k^{1/2} K^{-1}\right) \tag{3}$$

Due to space constraints, the proof of Theorem 1 is deferred to Appendix A.1.2.

*Insight* 1. Note that in Equation 3, for a fixed sparsity $k$, the estimator error guarantee decays with datapoints $n$ and RE constant $K$ while growing linearly with the tail $\beta_k$. Existing error guarantees in literature focus only on hard sparse $\boldsymbol{\theta}$ - the data matrix $\mathbf{X}$ satisfies $\mathsf{RE}(k, 3, \frac{\mathbf{X}}{\sqrt{n}})$ with constant $K'$ and $\gamma_2 = 0$ (see Theorem 7.13 Wainwright (2019)). However, with moderately stronger assumption of $\mathsf{RE}(k, 6, \frac{\mathbf{X}}{\sqrt{n}})$ on the data matrix, guarantees of Theorem 1 hold for all $\gamma_2 \leq 1/2$. As stated in Theorem 1, for a larger tail with $\gamma_2 > 1/2$, $\mathbf{X}$ needs to satisfy RE on a larger cone of vectors.

*Remark* 1. Note that the statistical guarantee presented in Theorem 1 is an offline error rate that is *robust to sparsity modeling assumption* - similar to Theorem 7.19 in Wainwright (2019) and Theorem 1.6 in Boche et al. (2015). However, the former holds only for the special case when $\mathbf{X}$ has i.i.d. Gaussian rows, and the latter requires the stronger RIP condition on the data matrix. Our error guarantee is much more general and holds for deterministic data matrices $\mathbf{X}$ satisfying RE.

Below we derive a corollary for the case when the rows of the design matrix are sampled without replacement from a set whose empirical covariance matrix has a minimum eigenvalue.

**Corollary 1.** *Let* $\mathbf{X} \in \mathbb{R}^{n \times d}$ *be the data matrix with* $n$ *samples and dimension* $d$, *whose rows are sampled uniformly without replacement from a set* $\mathcal{G} \subset \mathbb{R}^d$. *Let* $\Lambda = \lambda_{\min}(|\mathcal{G}|^{-1} \sum_{\mathbf{a} \in \mathcal{G}} \mathbf{a}\mathbf{a}^{\mathsf{T}})$. *Consider the same setup for observations* $\mathbf{r}$ *as in Theorem 1. Provided* $n = \Omega(k\Lambda^{-4})$, *an estimate* $\widehat{\boldsymbol{\theta}}$ *of* $\boldsymbol{\theta}$ *recovered using Lasso (Line 9 in* BSLB*), will satisfy with probability* $1 - \exp(-\Omega(n))$,

$$\|\boldsymbol{\theta} - \widehat{\boldsymbol{\theta}}\|_2 \leq \widetilde{O}\left(n^{-1/2} k^{1/2} \Lambda^{-1} + k^{-1/2} \beta_k + n^{-1/4} \beta_k^{1/2} \Lambda^{-1/2}\right). \tag{4}$$

Note in particular that we do not have the $(\gamma_2 \leq \gamma_1)$ assumption on the parameter vector $\boldsymbol{\theta}$ in Corollary 1. Instead, it is replaced by a lower bound on $n$ - datapoints sampled without replacement from the set $\mathcal{G}$ whose gram matrix has a sufficiently large minimum eigenvalue. This is possible because a lower bound on minimum eigenvalue for a positive semi-definite matrix implies a lower bound on RE with arbitrary parameters - concentration guarantees imply that the RE condition remains satisfied when sufficient (yet smaller than $|\mathcal{G}|$) number of datapoints are sampled from $\mathcal{G}$.

## 2.2 ONLINE GUARANTEES - REGRET BOUND FOR BSLB

Our next result is the expected regret incurred by BSLB in the online setting. The key ingredient in the regret analysis lies in appropriately accounting for the blocking constraint in the exploitation

component of BSLB. Below, we present the result detailing the regret guarantees of BSLB when the exploration period $T_{\text{explore}}$ is set optimally using a known sparsity level $k$. The proof is deferred to Appendix A.1.3 We invoke Corollary 1 at the end of the exploration component to obtain error guarantees of Lasso, which in turn bounds the maximum regret incurred in each step of the exploitation component. We optimize the exploration/exploitation trade-off to obtain our stated result.

**Theorem 2.** (***Regret Analysis of BSLB***) *Consider the $d$-dimensional sparse linear bandits framework with blocking constraint having a set $\mathcal{A} \subset \mathcal{B}^d$ of M arms spanning $\mathbb{R}^d$ and T rounds ($T \ll d \ll M$). In each round $t \in [T]$, we choose arm $\mathbf{a}_t \in \mathcal{A}$ and observe reward $r_t = \langle \boldsymbol{\theta}, \mathbf{a}_t \rangle + \eta_t$ where $\boldsymbol{\theta} \in \mathbb{R}^d$ is unknown and $\eta_t$ is zero-mean independent noise random variable with variance $\sigma^2 = O(1)$. Suppose $\boldsymbol{\theta}$ has tail magnitude $\beta_k := \|\boldsymbol{\theta}_{\mathcal{T}_k^c}\|_1$ at sparsity level $k$ where $\mathcal{T}_k \subseteq \{1, \ldots, d\}$ is the set of $k$ largest coordinates by absolute value. Let $\lambda_{\min}^*$ for the set $\mathcal{A}$ be as defined in equation 2 and assume that $\lambda_{\min}^* = \Omega(\log^2 M)$. In this framework, BSLB with exploration period $T_{\text{explore}} = \widetilde{O}(k^{\frac{1}{3}} T^{\frac{2}{3}})$, achieves a regret guarantee*

$$\mathbb{E}[\text{Reg}(T)] = \widetilde{O}\left(k^{\frac{1}{3}}(\lambda_{\min}^*)^{-1}T^{\frac{2}{3}} + k^{-\frac{1}{2}}\beta_k + k^{-\frac{1}{12}}(\lambda_{\min}^*)^{-1/2}\beta_k^{\frac{1}{2}}T^{\frac{5}{6}}\right). \tag{5}$$

*Insight* 2. BSLB enables diversity in selected arms by performing Step 2 in Alg. 1 - this step ensures that $\lambda_{\min}$ of the covariance matrix of the subset used in exploration is approximately optimal. The exploration period in Theorem 2 is optimized to maximize annotation of hard samples. However, in practice, the exploration period of BSLB can be increased further if diversity has more importance.

*Insight* 3. The runtime of the GETGOODSUBSET($\mathcal{A}, \hat{g}$) and the optimization in Step 9 of BSLB (LASSO) is Poly(M, $d$, T). However, if the mild assumption $\lambda_{\min}^* = \Omega(\log^2 M)$ not satisfied, then GETGOODSUBSET($\mathcal{A}, \hat{g}$) can be replaced with a (modified) Brute Force Algorithm that runs in time $O(M^d)$ (see Appendix A.1.6 for details), and the theorem statement still holds. Note that the stated runtime in the latter part is still significantly lower than the trivial brute force search for the optimal subset having a runtime of $O(\exp(M))$. This is possible because, as a result of the approximation guarantees of Theorem 3, we can restrict the size of the subset while performing a brute-force search.

For the case when the true parameter satisfies the hard sparsity condition (the tail $\beta_k$ is 0), our regret guarantee (see Corollary 2 in Appendix) achieves the same $T^{2/3}$ regret dependence as in as Hao et al. (2020) without the *blocking constraint*.

## 2.3 SUBSET SELECTION FOR MAXIMIZING THE MINIMUM EIGENVALUE

Recall that in Step 5 of BSLB; we sample from a carefully chosen subset of arms that has good coverage - more precisely, our goal is to solve the optimization problem in Eq. 2 to obtain a representative set of arms. Although Hao et al. (2020) had a similar objective, the absence of blocking constraint in their framework implied that they could solve for a probability distribution on the set of arms such that the minimum eigenvalue of the expected covariance matrix is maximized. Since their solution space was the probability simplex, the objective was continuous and concave - implying that a solution can be found efficiently. However, in our setting, due to the blocking constraint, we need to identify a subset of representative arms from which to sample uniformly at random without replacement in the exploration component - this leads to the objective in Eq. 2 being discrete and therefore non-convex. Note that a brute force solution to our objective implies a search over all subsets of [M] and will take time $\Omega(\exp(M))$. To design an efficient algorithm for obtaining a good feasible solution to the non-convex objective in 2, our first step is to obtain a convex relaxation as described in Eq. 6 - in particular, instead of optimizing over a subset, we optimize over probability distributions over the set of arms such that the probability mass over any arm is bounded from above.

$$\hat{\boldsymbol{\mu}}(\hat{g}) = \arg\max_{\boldsymbol{\mu} \in \mathcal{P}(\mathcal{A})} \lambda_{\min}\left(\mathbf{A}\text{diag}(\boldsymbol{\mu})\mathbf{A}^{\mathsf{T}}\right) \text{ such that } \|\boldsymbol{\mu}\|_\infty \leq \frac{1}{\hat{g}}, \tag{6}$$

Note that $\mathbf{A} = [\mathbf{a}^1, \ldots, \mathbf{a}^M]^{\mathsf{T}} \in \mathbb{R}^{M \times d}$ denotes the matrix with all arms and $\hat{g}$ is an additional parameter to the relaxed objective. Since the solution to Eq. 6 might not be a feasible one for Eq. 2, we use a randomized rounding (Step 19 in BSLB) procedure to obtain a feasible solution. In the randomized rounding procedure, each datapoint $j \in [M]$ is sampled into our feasible output set $\mathcal{G}$ (used in the exploration component) independently with probability $\hat{g}\boldsymbol{\mu}_j$.

Let $\mathcal{X} \subseteq \mathcal{A}$ with $g^* = |\mathcal{X}|$ be the optimal subset for which the RHS in Equation 2 is maximized and let $\lambda_{\min}^*$ be the corresponding objective value (minimum eigenvalue). We present the following theorem on the approximation guarantees of the solution achieved by our procedure GetGoodSubset

of Algorithm 1 - the theorem says that the minimum eigenvalue of the gram matrix associated with datapoints in $\mathcal{G}$ (obtained post randomized rounding procedure) is close to $\lambda_{\min}^*$.

**Theorem 3.** *Let* $\mathbf{A} = [\mathbf{a}^1, \ldots, \mathbf{a}^M]^\mathsf{T} \in \mathbb{R}^{M \times d}$ *denote the matrix of all arms. Consider the convex optimization of equation 6 solved at* $\hat{g} = \mathsf{O}(d)$. *Let* $\mathcal{G}$ *be the output of the randomized rounding procedure (Step 18-20 of Alg. 1) and* $\widehat{\lambda}_{\min}$ *be the minimum eigenvalue of the corresponding covariance matrix that is,* $\widehat{\lambda}_{\min} = \lambda_{\min}(|\mathcal{G}^{-1}| \sum_{\mathbf{a} \in \mathcal{G}} \mathbf{a}\mathbf{a}^\mathsf{T})$. *Then under the assumption* $\lambda_{\min}^* = \Omega((\log M)^2)$, *we must have* $\lambda_{\min}^* \leq 2\widehat{\lambda}_{\min}(\log M)^2$ *with probability* $1 - o(1)$.

If assumption of $\lambda_{\min}^* = \Omega((\log M)^2)$ is not satisfied in its place., then we can implement a brute-force search over all subsets whose size is in the range $[d, \alpha d]$ (for some constant $\alpha \geq 1$) to maximize the objective in Eq. 2 - note that the time complexity is still polynomial in the number of arms $M^{O(d)}$ (refer to Appendix A.1.6 for details) which is significantly improved than the trivial brute force algorithm which has a running time of $O(\exp M)$. Note that the modified brute-force algorithm enjoys stronger approximation guarantees, with $\frac{\lambda_{\min}^*}{2} \leq \widehat{\lambda}_{\min}$ and has a running time that is still polynomial in the number of arms $M$ but exponential in the dimension $d$.

*Remark* 2. We want to highlight that several existing techniques in experimental design deal with maximizing objectives such as minimum eigenvalue; however, existing work assumes submodularity or matroid constraints (Allen-Zhu et al., 2021), which the average minimum eigenvalue (normalized with size of the set) in our setting does not satisfy as discussed in Appendix A.1.7.

*Proof Outline:* We first show in Lemma 4 (using concentration guarantees) that the following two objective values are close namely (A) value of the maximized concave objective with distribution $\hat{\boldsymbol{\mu}} \in \mathcal{P}(\mathcal{A})$ and parameter $\hat{g}$ in Equation 6 (B) objective value of the set $\mathcal{G}$ (eq. 2) obtained via randomized rounding procedure from $\hat{\boldsymbol{\mu}}$ at $\hat{g}$ (line 19 in BSLB). Note that the value of the maximized concave objective in Equation 6 with parameter $g_1$ is larger than the value with parameter $g_2$ provided $g_1 \leq g_2$. Therefore, we show our approximation guarantees with respect to objective in Equation 6 with parameter $d$ - which in turn also translates into guarantees for the optimal parameter $g^*$ even though $g^*$ is unknown (since $g^* \geq d$). Finally, given that the concave objective with parameter $d$ in Equation 6 is a relaxation of the discrete objective in Equation 2, the objective value of the former is going to be larger than the objective value of the latter. Combining all these key ingredients, we have proved our theorem statement.

## 2.4 CORRALLING WHEN OPTIMAL SPARSITY LEVEL IS NOT KNOWN

Note that for any unknown parameter vector $\boldsymbol{\theta}$, we can fix the sparsity level $k$ and therefore the corresponding tail magnitude $\beta_k$ - subsequently, we can obtain the guarantees of Theorem 2 by setting the exploration period optimally for the fixed $k$. However, if $k$ is set too low, then $\beta_k$ will be too high, and therefore, the second term in regret (Equation 5) dominates. On the other hand, if $k$ is set too high, then $\beta_k$ is low but the first term in the regret bound dominates. There is a trade-off, and therefore, there is an optimal choice of sparsity $k^\star$ and, tail magnitude $\beta_{k^\star}$. Therefore, we propose a meta-algorithm C-BSLB that exploits coralling (Agarwal et al., 2017) multiple versions of the BSLB algorithm 1 with different values of $k$ used to set the exploration period $\mathsf{T}_{\text{explore}}$ - the meta-algorithm gradually learns to choose the best base algorithm. However, naively applying CORRAL with all distinct base algorithms leads to a linear dependence on dimension $d$ in the regret making it vacuous. Therefore we carefully choose $\log d$ base algorithms for search within CORRAL with corresponding sparsity parameters set on exponentially spaced points - such a restriction ensures that the overhead in regret is minimal (logarithmic dependence on dimension $d$). However, we still prove our regret guarantee with respect to the base algorithm with optimal sparsity - although it is not guaranteed that the optimal base algorithm will be in the set of carefully chosen base algorithms provided as input to the meta-algorithm.

**Theorem 4.** *Consider the* $d$-dimensional sparse linear bandits framework with blocking constraint as described in Theorem 2. Let the C-BSLB algorithm (Algorithm 3 in Appendix A.1.9) run with an appropriate learning rate on multiple versions of BSLB, using distinct sparsity parameter $k$ taking values in the set $\{2^i\}_{i=0}^{\lfloor \log_2(d) \rfloor + 1}$. Let the optimal sparsity parameter in Theorem 2 that achieves minimum regret be $k^\star \in \{1, 2, \ldots, d-1, d\}$, and let $\mathbb{E}[\mathsf{Reg}(\mathsf{T})]_*$ be the corresponding regret. Then the meta-algorithm C-BSLB achieves the following regret guarantee,

$$\mathbb{E}[\mathsf{Reg}(\mathsf{T})] = O(\sqrt{\mathsf{T} \log_2(d)} + \sqrt{k^\star} \log_2(d) \mathbb{E}[\mathsf{Reg}(\mathsf{T})]_*). \tag{7}$$

Note that the first term in Equation 7 and the multiplicative factor of $\sqrt{k^\star} \log_2(d)$ corresponds to the additional cost in combining the input base algorithms by Algorithm 3. We stress that the dependence on dimension $d$ from the additional cost is only logarithmic.

*Proof Outline:* We use Theorem 5 of (Agarwal et al., 2017) and our regret bound from Theorem 2 to obtain Theorem 4. The key novelty in our proof is to establish the following - when searching with the small curated set of base algorithms in CORRAL, we do not suffer a significant loss in the regret even if the base algorithm with the optimal sparsity parameter does not lie in the curated set. The crux of our proof lies in a covering argument. By using a recursive telescoping argument, we can bound the regret incurred between any base algorithm not used for the search while corraling versus the base algorithm with the closest sparsity parameter used in the search.

## 3 EXPERIMENTS

Below, we demonstrate our methods for annotation in a label-scarce setting for image classification on the PASCAL VOC 2012 dataset. Additional experimental results on SST-2 (text dataset) can be found in Appendix A.2.3. Finally, experiments on the MovieLens, Netflix, and GoodBooks datasets in the context of personalized recommendation with few labeled data using our theoretical framework are in Appendix A.2.1. Finally, we provide detailed simulations in Appendix A.2.4.

We consider the setting where we have a total of M unlabelled samples (with $\mathsf{T} \ll \mathsf{M}$) and only $\mathsf{T}$ datapoints can be annotated (sequentially). For each unlabeled data point sent for annotation to the expert(s), we receive the ground truth label and the *difficulty score* $r_t$ corresponding to the difficulty in annotating the data point. We showcase the effectiveness of BSLB (Alg. 1) in our experimental set-up with real-world datasets. Given a model $\mathcal{M}$ to be trained on a downstream task, to benchmark BSLB, we consider the following set of baselines (to compare against) to choose subset of datapoints for annotation and subsequent training of the aforementioned model $\mathcal{M}$:

1. **Random**: Subset of $\mathsf{T}$ unlabeled datapoints chosen uniformly at random
2. **All**: All the samples in training data (except the validation fold)
3. **AnchorAL** (Lesci & Vlachos, 2024): an anchoring based active learning baseline ($\mathsf{T}$ samples).
4. **SEALS** (Coleman et al., 2022): a KNN based sub-sampling active learning baseline ($\mathsf{T}$ samples).

$\tau_{\text{easy}}, \tau_{\text{hard}}$ (thresholds on difficulty score to determine easy/hard samples) and $\mathsf{T}_{\text{explore}}$ (exploration rounds) are relevant hyper-parameters specified for the corresponding experiments [2]. We benchmark learning performance on 2 datasets: a) $N_{\text{valid}}$ hard samples (samples with difficulty $> \tau_{\text{hard}}$) (**hard-valid**) b) $N_{\text{valid}}$ easy samples (samples with difficulty ratings $< \tau_{\text{easy}}$) (**easy-valid**).

**AnchorAL** and **SEALS** are state-of-the-art active learning (AL) algorithms. In general, for a label-scarce complex task, AL might not be immediately applicable (see cold-start problem in Li et al. (2024)) - especially for datapoints close to the decision boundary with noisy/less informative confidence intervals. This is because AL requires a reliably trained model on an even smaller subset of labeled datapoints - however, on datapoints far from the decision boundary (easy datapoints), noisy confidence signals are still useful. As we show in our experiment, this intuition holds, and the AL models, along with the **random** baseline, perform well on the **easy-valid** dataset. It is worth noting that complex (hard) datapoints often tend to be the main challenge in industrial applications. This is because it is *easy* to improve performance on easy data (cheaper to obtain) by simply increasing samples during training, but hard datapoints are difficult to generalize on (Pukowski & Lu, 2024).

**Image Classification on PASCAL VOC 2012**: Our main result is for the image classification task on the public image dataset, PASCAL VOC 2012 (Everingham et al., 2015). The dataset has $11,540$ unique images and comprises segmentations for 20 objects. In addition to the image dataset, we use difficulty scores of annotations from (Ionescu et al., 2016) - the authors have provided the visual search difficulty by measuring the time taken to annotate in a controlled environment. The annotation task here was to identify if an image contains a particular object, e.g. "Does this image contain a car". The authors derive a difficulty score between $0$ and $8$ by normalizing the time to annotate.

In our experiment, the goal is to train a learning model for image classification - $\mathcal{M}$ is a support vector machine (SVM) head attached to a frozen pre-trained vision transformer (ViT) model pre-trained on ImageNet-21k dataset (Wu et al., 2020). We present results on the classification task

---

[2]We consider the AL setup initialized with $\mathsf{T}_{\text{explore}}$ samples and $\mathsf{T} - \mathsf{T}_{\text{explore}}$ samples queried in a batch.

| Validation Type | Object Annotated | AnchorAL | SEALS | Random | All | **Our** (BSLB) |
|---|---|---|---|---|---|---|
| easy-valid | chair | $94.0 \pm 1.67$ | $90.6 \pm 1.8$ | $96.4 \pm 1.0$ | $\mathbf{96.0 \pm 1.1}$ | $94.6 \pm 1.6$ |
| | car | $94.5 \pm 1.6$ | $94.7 \pm 4.0$ | $97.7 \pm 2.1$ | $\mathbf{98.7 \pm 0.1}$ | $96.5 \pm 1.8$ |
| | bottle | $93.0 \pm 2.5$ | $92.8 \pm 2.3$ | $96.8 \pm 1.1$ | $\mathbf{96.8 \pm 1.1}$ | $94.8 \pm 2.0$ |
| | bottle or chair | $91.5 \pm 1.1$ | $92.3 \pm 1.1$ | $94.8 \pm 0.97$ | $\mathbf{94.6 \pm 2.1}$ | $91.7 \pm 2.2$ |
| **hard-valid** | chair | $69.3 \pm 3.1$ | $69.6 \pm 6.1$ | $66.0 \pm 3.8$ | $71.3 \pm 3.2$ | $\mathbf{73.3 \pm 3.3}$ |
| | car | $70.3 \pm 4.0$ | $70.0 \pm 5.7$ | $60.0 \pm 5.4$ | $65.4 \pm 4.0$ | $\mathbf{74.0 \pm 3.4}$ |
| | bottle | $63.1 \pm 2.9$ | $63.4 \pm 3.4$ | $59.7 \pm 4.4$ | $64.8 \pm 1.9$ | $\mathbf{66.8 \pm 2.6}$ |
| | bottle or chair | $67.1 \pm 3.5$ | $66.3 \pm 1.4$ | $68.0 \pm 4.0$ | $72.3 \pm 2.0$ | $\mathbf{73.0 \pm 1.7}$ |

Table 1: Test accuracy of model $\mathcal{M}$ trained on different subsets of data annotated for 4 distinct object detection tasks in an image (PASCAL-VOC): The test performance of BSLB approach on the easy and hard validation dataset is at par with the $\mathcal{M}$ trained on all samples. We perform significantly better on the hard validation dataset compared to random sampling and active learning baselines.

- given an input image, predict if the image has an *object* or not. We consider 4 different objects, namely chair, car, bottle, and (bottle or chair). The last object is an OR conjunction of two labels. We consider the thresholds as $\tau_{\text{easy}} = 3.1$ and $\tau_{\text{hard}} = 3.9$ since the distribution of the difficulty scores in the dataset is heavy-tailed as shown in Figure 4b. The (image, question) tuple with difficulty scores in the range $[3.1, 3.8]$ are highly noisy and therefore have been excluded. Table 2 contains the hyperparameters $\mathsf{T}$, $\mathsf{T}_{\text{explore}} (\approx 0.6\mathsf{T})$ used and the number of samples in the **all** dataset for the different object classification tasks, along with the size of the validation datasets **hard-valid** and **easy-valid** and aggregaged accuracies. Table 3 contains results on the effect of varying $\mathsf{T}_{\text{explore}}$.

We present our results in Table 1 averaged over 5 validation folds. For this classification task, our method (BSLB) efficiently selects datapoints (to be annotated) compared to baselines with an equal number of samples. Regarding the quality of the final trained model $\mathcal{M}$, the learning performance of BSLB on **easy-valid** is within $2\%$ of that obtained by the baseline **random**. However, there is an improvement of $5 - 14\%$ on the hard validation data **hard-valid**. When compared to the active learning baselines (**AnchorAL** and **SEALS**), BSLB performs better by $1 - 4\%$ on **easy-valid** and by $3.5 - 7\%$ on **hard-valid**. Finally, when compared to $\mathcal{M}$ trained on all datapoints (**all** baseline), which has $6\times$ to $12\times$ more samples, our method does better ($0.7\%$ to $8.6\%$) on the **hard-valid** and does decently on **easy-valid** ($< 3\%$ difference). These results validate our theory - in particular, we find that performance on **easy-valid** improves if the model $\mathcal{M}$ is trained on more samples (randomly chosen to improve coverage). However, improving the performance on **hard-valid** dataset is the main challenge where our simple approach BSLB with theoretical guarantees does reasonably well.

## 4 CONCLUSION AND FUTURE WORK

This work formulates the problem of adaptive annotation with expert-provided difficulty feedback as regret minimization in blocked sparse linear bandits. The goal is to annotate difficult samples to improve generalization on hard datapoints, assuming difficulty ratings are linked to sample features by an unknown parameter. We consider the practical consideration when each sample can be annotated once and only a few rounds of annotations are available. We propose an explore-then-commit algorithm BSLB, a two-shot algorithm for unlabelled data-point selection (for annotation). We show theoretical results establishing the sub-linear regret of the proposed BSLB algorithm - we also ensure that our guarantees work with minimum assumptions on datapoints and are robust to sparsity. Finally, we present a meta-algorithm C-BSLB that corrals BSLB algorithms with different input parameters - the result is a bandit algorithm that suffers the same regret orders as BSLB without the knowledge of the optimal sparsity hyper-parameter. Numerical studies on real-life image and text datasets show the efficacy of our methods in the label-scarce regime. We demonstrate experiments in the appendix for recommendation and simulation setup using our theoretical framework and BSLB.

**Future Work:** Our experimental results are done on classifying state-of-the-art embeddings; however, more extensive experimentation can be considered in future work, especially for creating datasets used for generative tasks.

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

# A  APPENDIX

The Appendix comprises two sections: Section A.1 details the detailed proofs of the technical results and Section A.2 presents additional numerical results.

## A.1  BRIEF OVERVIEW AND TECHNICAL PROOFS

### A.1.1  LOWER BOUND FOR REGRET

We provide the following theorem on the lower bound of regret for the high-dimensional linear bandit setting with blocking constraint. The following lower bound is derived for a parameter vector with hard sparsity but trivially extends to a parameter vector with soft sparsity.

**Theorem 5.** *Consider the $d$-dimensional sparse linear bandits framework with blocking constraint having a set $\mathcal{A} \subseteq \mathcal{B}^d$ of M arms spanning $\mathbb{R}^d$ and T rounds ($\mathsf{T} \ll d \ll \mathsf{M}$). In each round $t \in [\mathsf{T}]$, we choose arm $a_t \in \mathcal{A}$ and observe reward $r_t = \langle \theta, a_t \rangle + \eta_t$ where $\theta \in \mathbb{R}^d$, is unknown and $\eta_t$ is zero-mean independent noise random variable with variance $\sigma^2 = 1$. Assume that the parameter vector is $k$-sparse, $\|\theta\|_0 = k$. Then for any bandit algorithm the worst case regret is lower bounded as follows,*

$$\mathbb{E}[\mathsf{R}] = \Omega(\min(k^{1/3}\mathsf{T}^{2/3}, \sqrt{d\mathsf{T}})).$$

*Proof.* We consider the hard-sparsity instance of the high-dimensional linear bandit setting with the *blocking* constraint. We prove this in three steps. First we show how one can construct an equivalent bandit problem with a *blocking* constraint for any problem without the *blocking* constraint. Next we use results from Hao et al. (2020) to show that the result holds true for this transformation. Finally, we show that this is the best

1. For any instance of the bandit problem without the *blocking* constraint, we can construct a bandit problem with the *blocking* constraint. This can be done by considering T copies of each of the arms, i.e. for the original arm set $\mathcal{A} = \{a^{(1)}, \ldots, a^{(\mathsf{M})}\}$ we construct the arm multiset $\mathcal{A}'$ as follows,

$$\mathcal{A}' = \cup_{i=1}^{\mathsf{M}} \cup_j^{\mathsf{T}} \{a_j^{(i)}\}$$

   where $a_j^{(i)}$ denotes the $j^{\text{th}}$ copy of the $i^{\text{th}}$ arm (such that it is the different arm with the same arm vector). Now the bandit setting with blocking constraint with arm set $\mathcal{A}'$ is identical to the bandit setting without blocking constraint with arm set $\mathcal{A}$. Further the regret decomposition becomes identical in the first term, i.e. the top T arms have the same arm vector.

2. Now for any arm set $\mathcal{A}$ without the blocking constraint, the lower bound from Theorem 3.3 of Hao et al. (2020) holds for any algorithm, and the following bound holds

$$\mathbb{E}[\mathsf{R}] = \Omega(\min(k^{1/3}\mathsf{T}^{2/3}, \sqrt{d\mathsf{T}})).$$

3. Now if any algorithm operating with the blocking constraint could achieve the regret of order lesser than $\min(k^{1/3}\mathsf{T}^{2/3}), \sqrt{d\mathsf{T}}$, then the algorithm would solve the bandit problem with arm multiset $\mathcal{A}'$ (with the blocking constraint) with regret lower than $\min(k^{1/3}\mathsf{T}^{2/3}), \sqrt{d\mathsf{T}}$. But then we can solve the original problem with arm set $\mathcal{A}$ with the same regret, hence arriving at a contradiction.

In the data-poor regime $d \geq k^{1/3}\mathsf{T}^{2/3}$, which is the regime considered in the paper, this bound reduces to $\Omega(k^{1/3}\mathsf{T}^{2/3})$, which is the order that our upper bound achieves.

$\square$

### A.1.2 PROOF OF THEOREM 1

We defined the restricted eigenvalue of a matrix as,

**Definition 2.** *Restricted Eigenvalue: If $\mathbf{X}$ satisfies Restricted Eigenvalue property $\mathsf{RE}(k_0, \gamma, \mathbf{X})$, then there exists a constant $K(k_0, \gamma, \mathbf{X})$ such that for all $z \in \mathbb{R}^d$ and $z \neq \mathbf{0}$,*

$$K(k_0, \gamma, \mathbf{X}) = \min_{J \subseteq \{1,\ldots,d\}, |J| \leq k_0} \min_{\|z_{J^c}\|_1 \leq \gamma\|z_J\|_1} \frac{\|\mathbf{X}z\|_2}{\|z_J\|_2}$$

This definition implies,

$$K(k_0, \gamma, \mathbf{X})\|z_J\|_2 \leq \|\mathbf{X}z\|_2 \; \forall \|z_{J^c}\|_1 \leq \gamma\|z_J\|_1 \; \forall J \subseteq \{1,\ldots,d\}, |J| \leq k_0$$

We prove the theorem by using the Basis Pursuit Program which is one of three equivalent formulation for LASSO Chen et al. (2001); Wainwright (2019),

$$\widehat{\boldsymbol{\theta}} = \arg\min_{\boldsymbol{\theta}} \|\boldsymbol{\theta}\|_1$$

$$\text{s.t.} \sum_{t \in [\mathsf{T}_{\text{explore}}]} (\mathcal{R}[t] - \langle \boldsymbol{\theta}, \mathcal{C}[t] \rangle)^2 \leq \lambda \tag{8}$$

*Proof.* For the purpose of the proof denote the true parameter by $\boldsymbol{\theta}^*$. The following inequality holds for a general basis pursuit program (w/o any assumptions on sparsity) Wainwright (2019),

$$\frac{1}{n}\|\mathbf{Xh}\|_2^2 \leq 2\frac{\mathbf{w}^\top \mathbf{Xh}}{n} + 2(b^2 - \frac{\|\mathbf{w}\|_2^2}{2n}) \tag{9}$$

where $\mathbf{h} = \widehat{\boldsymbol{\theta}} - \boldsymbol{\theta}^*$, $\mathbf{w} = \mathcal{R} - \mathbf{X}\boldsymbol{\theta}^*$, $b^2$ is the tolerance and $n$ is the number of samples . For the proof, the tolerance is equal to the regularization parameter $b^2 = \lambda$.

Let $A = 2(b^2 - \frac{\|\mathbf{w}\|_2^2}{2n})$ $B = \|\frac{\mathbf{Xw}}{n}\|_\infty$ and $\alpha = \|\mathbf{h}_{(\mathcal{T}_0 \cup \mathcal{T}_1)}\|_2$, where $\mathcal{T}_0$ denotes the top-$k$ coordinates, $\mathcal{T}_1$ the next top $k$ coordinates and so on, for this proof.

First note that,

$$\|\mathbf{h}_{(\mathcal{T}_0 \cup \mathcal{T}_1)^c}\|_1 = \sum_{m \geq 2} \|\mathbf{h}_{\mathcal{T}_m}\|_1 \leq \sum_{m \geq 2} k \max_{j \in \mathcal{T}_m} \mathbf{h}_j \leq \sum_{m \geq 2} k \min_{j \in \mathcal{T}_{m-1}} \mathbf{h}_j \leq \sum_{m \geq 1} \|\mathbf{h}_{\mathcal{T}_m}\|_1 \leq \|\mathbf{h}_{(\mathcal{T}_0)^c}\|_1$$

And from the decomposition available in Theorem 1.6 in Boche et al. (2015) we know that

$$\|\mathbf{h}_{(\mathcal{T}_0)^c}\|_1 \leq \|\mathbf{h}_{(\mathcal{T}_0)}\|_1 + 2\beta_k$$

We also the following inequality,

$$\|\boldsymbol{\theta}^*_{\mathcal{T}_0}\|_1 = \|\boldsymbol{\theta}^*_{\mathcal{T}_0}\|_1 - \|\widehat{\boldsymbol{\theta}}_{\mathcal{T}_0}\|_1 + \|\widehat{\boldsymbol{\theta}}_{\mathcal{T}_0}\|_1 \leq \|\boldsymbol{\theta}^*_{\mathcal{T}_0} - \widehat{\boldsymbol{\theta}}_{\mathcal{T}_0}\|_1 + \|\widehat{\boldsymbol{\theta}}_{\mathcal{T}_0}\|_1 = \|\mathbf{h}_{\mathcal{T}_0}\|_1 + \|\widehat{\boldsymbol{\theta}}_{\mathcal{T}_0}\|_1 \leq 2\|\mathbf{h}_{\mathcal{T}_0}\|_1$$

The last inequality holds since $\widehat{\boldsymbol{\theta}}$ is a solution of equation 8 and $\|\widehat{\boldsymbol{\theta}}_{\mathcal{T}_0}\|_1 \leq \|\widehat{\boldsymbol{\theta}}_{\mathcal{T}_0}\|_1 \leq \|\mathbf{h}_{\mathcal{T}_0}\|_1$ otherwise $\mathbf{h}_{\mathcal{T}_0}$ would be the solution instead.

Therefore,

$$\|\mathbf{h}_{\mathcal{T}_0^c}\|_1 \leq \|\mathbf{h}_{\mathcal{T}_1}\|_1 + \|\mathbf{h}_{(\mathcal{T}_0 \cup \mathcal{T}_1)^c}\|_1 = \|\mathbf{h}_{\mathcal{T}_1}\|_1 + \|\mathbf{h}_{(\mathcal{T}_0 \cup \mathcal{T}_1)}\|_1 + 2\beta_k$$
$$\leq \|\mathbf{h}_{\mathcal{T}_1}\|_1 + \|\mathbf{h}_{(\mathcal{T}_0 \cup \mathcal{T}_1)}\|_1 + 2\gamma\|\boldsymbol{\theta}^*_{\mathcal{T}_0}\|_1 \leq 2\|\mathbf{h}_{\mathcal{T}_0}\|_1 + 2\|\mathbf{h}_{\mathcal{T}_0}\|_1 + 4\gamma\|\mathbf{h}_{\mathcal{T}_0}\|_1 \leq 4(1+\gamma)\|\mathbf{h}_{\mathcal{T}_0}\|_1$$

Now by the RE condition,

$$\frac{\|\mathbf{Xh}\|_2^2}{n} \geq K^2(k_0, 4+4\gamma, \frac{\mathbf{X}}{\sqrt{n}})\|\mathbf{h}_{\mathcal{T}_0 \cup \mathcal{T}_1}\|_2^2$$

Putting this into equation 9,

$$K^2\|\mathbf{h}_{\mathcal{T}_0}\|_2^2 \leq 2\|\mathbf{h}\|_1\|\frac{\mathbf{X}^\top \mathbf{w}}{n}\|_\infty + 2(b^2 - \frac{\|\mathbf{w}\|_2^2}{2n}) \leq B\|\mathbf{h}\|_1 + A$$

We now bound $\|\mathbf{h}\|_1$

$$\|\mathbf{h}\|_1 = \|\mathbf{h}_{(\mathcal{T}_0)}\|_1 + \|\mathbf{h}_{(\mathcal{T}_0)^c}\|_1$$
$$\|\mathbf{h}_{(\mathcal{T}_0)}\|_1 \leq \sqrt{k}\|\mathbf{h}_{(\mathcal{T}_0)}\|_2$$

And the bound on $\|\mathbf{h}_{(\mathcal{T}_0)^c}\|_1$ is,

$$\|\mathbf{h}_{(\mathcal{T}_0)^c}\|_1 \leq \|\mathbf{h}_{(\mathcal{T}_1)}\|_1 + \|\mathbf{h}_{(\mathcal{T}_0 \cup \mathcal{T}_1)^c}\|_1 \leq \|\mathbf{h}_{(\mathcal{T}_1)}\|_1 + \|\mathbf{h}_{(\mathcal{T}_0 \cup \mathcal{T}_1)}\|_1 + 2\beta_k(\boldsymbol{\theta}^*)_1 \leq 4\sqrt{k}\|\mathbf{h}_{(\mathcal{T}_0)}\|_2 + 2\beta_k(\boldsymbol{\theta}^*)_1$$

Combining both bounds we find the bound on $\|\mathbf{h}\|_1$,

$$\|\mathbf{h}\|_1 \leq 5\sqrt{k}\|\mathbf{h}_{(\mathcal{T}_0)}\|_2 + 2\beta_k(\boldsymbol{\theta}^*)_1 = C\alpha + 2\beta$$

where we make the following substitution ($C = 5\sqrt{k}, \beta = \beta_k(\boldsymbol{\theta}^*)_1$)

$$K^2\alpha^2 \leq B(C\alpha + 2\beta) + A \implies K^2\alpha^2 \leq (BC)\alpha + A + 2B\beta$$

Let $F = K^2(k_0, 4 + 4\gamma, \frac{1}{\sqrt{n}}\mathbf{X})$,

$$\alpha^2 \leq \frac{BC}{F}\alpha + \frac{A + 2B\beta}{F}$$

This is true when,

$$\alpha \leq 2\frac{BC}{F} + \sqrt{\frac{A + 2B\beta}{F}}$$

Now applying concentration using the properties of a bounded zero-mean noise, $B \leq C_1\sigma(\sqrt{\frac{2\log d}{n}} + \epsilon)$ w.h.p. and $A \leq \sigma^2\epsilon$ w.h.p. (Denote the union bound on the probability of both these events by $1 - 3\exp(-\zeta_1 n)$ for some constant $\zeta_1$ dependent on $\epsilon$ and other constants),

$$||\mathbf{h}||_2 \leq ||\mathbf{h}_{(\mathcal{T}_0)}||_2 + ||\mathbf{h}_{(\mathcal{T}_0)^c}||_2 \leq 4\frac{(BC)}{F} + \sqrt{\frac{A + 2B\beta}{F}} + \frac{1}{\sqrt{k}}\beta$$

$$||\mathbf{h}||_2 \leq 4\frac{BC}{F} + \sqrt{\frac{A + 2B\beta}{F}} + \frac{1}{\sqrt{k}}\beta$$

$$||\mathbf{h}||_2 \leq 4\frac{(C_1\sigma(\sqrt{\frac{2\log d}{n}} + \epsilon)(5\sqrt{k}))}{F} + \sqrt{\frac{\sigma^2\epsilon + 2C_1\sigma(\sqrt{\frac{2\log d}{n}} + \epsilon)\beta}{F}} + \frac{1}{\sqrt{k}}\beta$$

with a high probability $1 - 3\exp(-\zeta_2 n)$.

We simplify the above expression in terms of the number of samples $n$, sparsity parameter $k$, the tail parameter $\beta$ and the RE constant.

$$||\mathbf{h}||_2 \leq \zeta_3 n^{-1/2}k^{1/2}K^{-2} + \zeta_4 k^{-1/2}\beta + \zeta_5 n^{-1/4}\beta^{1/2}K^{-1}$$

$$||\mathbf{h}||_2 \leq \mathsf{O}\left(n^{-1/2}k^{1/2}K^{-2} + k^{-1/2}\beta + n^{-1/4}\beta^{1/2}K^{-1}\right)$$

$\square$

### A.1.3 PROOF OF COROLLARY 1

The only additional ingredient needed to prove the corollary is the following concentration inequality, which shows that if the RE of the covariance matrix of the set which is used for sampling is bounded from below, then so is the RE of the sampled covariance matrix with high probability. Since in the assumption of the corollary, the minimum eigenvalue of the covariance matrix of the set used for sampling is $> 0$, and the RE is bounded from below by the minimum eigenvalue, the corollary follows.

However although the below theorem is adapted directly from Rudelson & Zhou (2013); to prove it we need to make changes accordingly for the sampling without replacement case. This introduces an additional multiplicative $\lambda_{\min}$ term in the exponential in the probability but does not change the order with respect to any other variable. We now

*Concentration Inequality for RE condition to hold on sample covariance matrix* We use the following Theorem from Rudelson & Zhou (2013).

**Theorem 6.** *Let $0 < \delta < 1$ and $0 < k_0^\star < d$. Let $Y \in \mathbb{R}^d$ be a random vector such that $\|Y\|_\infty \leq M$ a.s and denote $\Sigma = \mathbb{E}YY^T$. Let $\mathbf{X}$ be an $n \times d$ matrix, whose rows $X_1, \ldots, X_n$ are sampled without replacement from a set $\mathcal{G}$. Let $\Sigma$ satisfy the $\mathsf{RE}(k_0^\star, 3k_0, \Sigma^{1/2})$ condition as in Definition 1. Set $k_1^\star = k_0^\star + k_0^\star \max_j \|\Sigma^{1/2}e_j\|_2^2 \times \left(\frac{16\mathsf{RE}(k_0^\star, 3\Lambda_0, \Sigma^{1/2})^2(3\Lambda_0)^2(3\Lambda_0 + 1)}{\delta^2}\right)$. Assume that $k_1^\star \leq d$ and $\rho = \rho_{\min}(d, \Sigma^{1/2}) > 0$. Suppose the sample size satisfies for some absolute constant $C$*

$$n \geq \frac{CM^2 k_1^\star \cdot \log d}{\rho\delta^2} \cdot \log^3\left(\frac{CM^2 k_1^\star \cdot \log d}{\rho\delta^2}\right).$$

*Then, with probability at least* $1 - \exp(-\delta\rho^2 n/(6M^2 d))$, $\mathsf{RE}(k_0^\star, k_0, \mathbf{X})$ *condition holds for matrix* $\frac{1}{\sqrt{n}}\mathbf{X}$ *with*

$$0 \geq (1 - \delta)K(k_0^\star, k_0, \Sigma^{1/2}) \leq K\left(k_0^\star, k_0, \frac{1}{\sqrt{n}}\mathbf{X}\right).$$

(The inequality is reverse because our definition of Restricted Eigenvalue has the $1/K$ compared to the definition of Rudelson & Zhou (2013) )

The proof of Theorem 6 is dependent on Theorem 23, which is reproduced below, and Theorem 10 (Reduction Principle), which is in the paper. Out of these two, only Theorem 23 has elements related to the randomness of the design. In the proof of the above theorem, authors use concentration inequality to extend a RIP-like condition on a general cone (rather than sparse vectors). This concentration inequality results from the following theorem: an augmented version of Theorem 22 of Rudelson & Zhou (2013) to the sampling without replacement case. *The original proof of Theorem 22 is extremely involved (and mathematically rich). Reproducing the entire proof would have surmounted to reproducing the entire paper. We only highlight the key difference, it is recommended that the reader goes through the proof beforehand/side-by-side.*

**Theorem 7.** *Set* $1 > \delta > 0$, $0 < k_0^\star \leq d$ *and* $\Lambda_0 > 0$. *Let* $\mathcal{G}$ *be a subset of vectors such that* $||Y||_\infty \leq M$, *with* $\Sigma = \sum_{Y \in \mathcal{G}} \frac{1}{|\mathcal{G}|}YY^\mathsf{T}$. $\Sigma$ *satisfies* $\mathsf{RE}(k_0^\star, 3\Lambda_0, \Sigma^{1/2})$. *Rows of* $\mathbf{X}$ *are drawn uniformly without replacement from* $\mathcal{G}$. *Set* $k_1^\star = k_0^\star + k_0^\star \max_j \|\Sigma^{1/2}e_j\|_2^2 \times$ $\left(\frac{16\mathsf{RE}(k_0^\star, 3\Lambda_0, \Sigma^{1/2})^2 (3\Lambda_0)^2 (3\Lambda_0 + 1)}{\delta^2}\right)$. *Assume* $k_1^\star \leq d$ *and* $\lambda_{\min}(k_1^\star, \Sigma^{1/2}) > 0$. *If for some absolute constant* $\zeta_{12}$,

$$n \geq \frac{\zeta_{12}M^2 k_1^\star \log d}{\lambda_{\min}(k_1^\star, \Sigma^{1/2})\delta^2} \log^3\left(\frac{\zeta_{12}M^2 k_1^\star \log d}{\lambda_{\min}(k_1^\star, \Sigma^{1/2})\delta^2}\right)$$

*then with probability* $1 - \exp(\frac{-\delta\lambda_{\min}^2(k_1^\star, \Sigma^{1/2})n}{6M^2 k_1^\star})$, *for all* $\mathbf{v} \in \mathsf{C}(k_0^\star, \Lambda_0)$, $\mathbf{v} \neq 0$

$$1 - \delta \leq \frac{1}{\sqrt{n}}\frac{\|\mathbf{X}\mathbf{v}\|_2}{\|\mathbf{v}\|_2} \leq 1 + \delta.$$

*Proof.* In the proof of Theorem 22 of Rudelson & Zhou (2013), two arguments require the sampling with replacement (i.i.d. samples), namely symmetrization and Talagrands concentration inequality. We use the sampling without a replacement version of McDiarmid's concentration inequality to obtain comparable bounds. Therefore, to prove this argument, the following two lemmas.

**Lemma 1.** *(Symmetrization without Replacement)*

$$\mathbb{E}\sup_{x \in F}\left|\mathbb{E}f_j(x, Z_j) - \frac{1}{n}\sum f_j(x, Z_j)\right| \leq \frac{2}{n}\mathbb{E}\sup_{x \in F}\left|\sum \xi_j f_j(x, Z_j)\right|$$

*where are i.i.d. Rademacher random variables and* $Z_j$ *are random variables sampled uniformly without replacement from some set.*

*Proof.* Let $Z_1, \ldots, Z_n$ be the random variables sampled uniformly without replacement from set $\mathcal{G}$. $Z_1-$ Consider $Z_1^{'}, \ldots, Z_n^{'}$ be an independent sequence of random variables sampled uniformly without replacement from set $\mathcal{G}$. Then $\frac{1}{n}\sum f_j(x, Z_j) - \mathbb{E}f_j(x, Z_j)$ and $\frac{1}{n}\sum f_j(x, Z_j^{'}) - \mathbb{E}f_j(x, Z_j)$ are zero mean random variable. Then,

$$\mathbb{E}\|\frac{1}{n}\sum f_j(x, Z_j) - \mathbb{E}f_j(x, Z_j)\| \leq \mathbb{E}\|\frac{1}{n}\sum f_j(x, Z_j) - \mathbb{E}f_j(x, Z_j) - \frac{1}{n}\sum f_j(x, Z_j^{'}) + \mathbb{E}f_j(x, Z_j)\|$$

$$\implies \mathbb{E}\|\frac{1}{n}\sum f_j(x, Z_j) - \mathbb{E}f_j(x, Z_j)\| \leq \mathbb{E}\|\frac{1}{n}\sum \left(f_j(x, Z_j) - f_j(x, Z_j^{'})\right)\|$$

(Since $\frac{1}{n}\sum f_j(x, Z_j) - \mathbb{E}f_j(x, Z_j)$ and $\frac{1}{n}\sum f_j(x, Z_j^{'}) - \mathbb{E}f_j(x, Z_j)$ are independent)

$$\mathbb{E}\|\frac{1}{n}\sum f_j(x, Z_j) - \mathbb{E}f_j(x, Z_j)\| \leq \mathbb{E}\|\frac{1}{n}\sum \left(f_j(x, Z_j) - f_j(x, Z_j^{'})\right)\|$$

$$\implies \mathbb{E}\|\frac{1}{n}\sum \xi_j \left(f_j(x, Z_j) - f_j(x, Z_j^{'})\right)\| \leq 2\mathbb{E}\|\frac{1}{n}\sum \xi_j \left(f_j(x, Z_j)\right)\|$$

(Symmetric random variables (Vershynin, 2018) since $Z_j$ and $Z_j'$ have same distribution; followed by Triangular Inequality). $\qquad\square$

**Lemma 2.** *(Concentration using McDiarmid's inequality) If $|f_j(x)| \leq \zeta_{13}$ a.s.. And suppose $W = \sup_{x \in F} \sum_{j=1}^{n} f_j(x, Z_j)$, where $Z_1, \ldots, Z_n$ are sampled uniformly without replacement from some set. If $\mathbb{E} W \leq 2\delta n$.*

$$\mathbb{P}(W \geq 4\delta n) \leq \exp\left(\frac{-8\delta^2 n}{\zeta_{13}^2}\right)$$

*Proof.* We prove the result by using McDiarmid's inequality. First we bound the quantity,

$$\sup_{Z_i'} |\sup_{x \in F} \sum_{j=1}^{n} f_j(x, Z_j) - \sup_{x \in F}\left(\sum_{j=1, j \neq i}^{n} f_j(x, Z_j) + f_i(x, Z_i')\right)|$$

$$\leq \sup_{Z_i'} |\sup_{x \in F} \sum_{j=1, j \neq i}^{n} f_j(x, Z_j) + \sup_{x \in F} f_j(x, Z_i) - \sup_{x \in F} \sum_{j=1, j \neq i}^{n} f_j(x, Z_j) - \inf_{x \in F} f_i(x, Z_i')|$$

$$\leq \sup_{Z_i'} |\sup_{x \in F} f_j(x, Z_i)| \leq \zeta_{13}$$

We use the loose version of McDiarmid's concentration inequality for random variable without replacement with $t = 2\delta n$ to obtain the result. The condition that needs to be verified is that $W$ is symmetric under permutations of the individual $f_j(x, Z_j)$. This is obviously true. We next state McDiarmid's concentration inequality without replacement from Tolstikhin (2017),

**Lemma 3.** *Suppose $W = \sup_{x \in F} \sum_{j=1}^{n} f_j(x, Z_j)$, where $Z_1, \ldots, Z_n$ are sampled uniformly without replacement from some set. Then,*

$$\mathbb{P}(W - \mathbb{E} W \geq t) \leq \exp\left(\frac{-2t^2}{n\zeta_{13}^2}\right).$$

The probability is $\exp\left(\frac{-8\delta^2 n}{\zeta_{13}^2}\right) \leq \exp\left(\frac{-\delta^2 n}{6\zeta_{13}^2}\right) \leq \exp\left(\frac{-\delta^2 n \lambda_{\min}^2(k_1^\star, \Sigma^{1/2})}{6M^4 m^2}\right) \leq \exp\left(\frac{-\delta^2 n \lambda_{\min}^2(k_1^\star, \Sigma^{1/2})}{6M^2 m}\right)$, which is same as that in the original theorem except an additional $\lambda_{\min}(k_1^\star, \Sigma^{1/2})$ which is reflected in the Theorem statement. $\qquad\square$

*Comment on Dudley's inequality*: Theorem 23 also uses Dudley's inequality, but there the $\Psi_1, \ldots, \Psi_n$ are treated as deterministic and so the proof goes through in our sampled without replacement case as well. $\qquad\square$

We can now complete the proof by computing the probability of the following event,

$$\mathcal{E} = \{\mathsf{RE}(k, 4 + 4\gamma, \hat{\Sigma}) \text{ constant } K \geq {\lambda_{\min}^*}^{1/2},$$
$$\|\boldsymbol{\theta} - \widehat{\boldsymbol{\theta}}\|_2 = \widetilde{\mathrm{O}}\left(\mathsf{T}_{\text{explore}}^{-1/2} k^{1/2} K^{-2} + k^{-1/2}\beta + \mathsf{T}_{\text{explore}}^{-1/4} \beta^{1/2} K^{-1}\right)\}$$

has the probability, $\mathbb{P}(\mathcal{E}) \geq 1 - 3\exp(-\zeta_5 \mathsf{T}_{\text{explore}})$ which completes the proof.

### A.1.4 PROOF OF THEOREM 2

*Proof.* The regret bound can be proved in 3 steps. First, we decompose the regret, apply Corollary 1, and then optimize the exploration period.

**Step 1. Regret Decomposition:** Define the maximum reward as, $R_{\max} = \max_{\mathbf{a} \in \mathcal{A}} \boldsymbol{\theta}^{\mathsf{T}} \mathbf{a}$ and $\mathbf{a}^*$ as the corresponding arm. As shown in the Appendix, the regret can be decomposed as, this step requires care since the regret is with respect to the top-$\mathsf{T}_{\text{explore}}$ arms. In the exploitation stage, the arms are selected such that the top $(\mathsf{T} - \mathsf{T}_{\text{explore}})$−arms are played according to $\widehat{\boldsymbol{\theta}}$ and are indexed by the permutation $\widehat{\pi}$. We next bound the regret for the $j^{\text{th}}$ selected arm.

1. If $\boldsymbol{\theta}^{\mathsf{T}}\mathbf{a}^{(\widehat{\pi}(j))} \geq \boldsymbol{\theta}^{\mathsf{T}}\mathbf{a}^{(\pi(j))}$, then the regret for the $j^{\text{th}}$ selected arm is negative.

2. If not i.e., $\boldsymbol{\theta}^{\mathsf{T}}\mathbf{a}^{(\widehat{\pi}(j))} \leq \boldsymbol{\theta}^{\mathsf{T}}\mathbf{a}^{(\pi(j))}$, then there exists an arm index, $j_1$ in the permutation $\pi$ such that $j_1$ is shifted to the left in $\widehat{\pi}$. This implies that $\boldsymbol{\theta}^{\mathsf{T}}\mathbf{a}^{(\widehat{\pi}(j_1))} \geq \boldsymbol{\theta}^{\mathsf{T}}\mathbf{a}^{(\pi(j_1))}$. We decompose the regret for this case with respect to this index and bound the error:

$$\boldsymbol{\theta}^{\mathsf{T}}\mathbf{a}^{(\pi(j))} - \boldsymbol{\theta}^{\mathsf{T}}\mathbf{a}^{(\widehat{\pi}(j))} = \underbrace{(\boldsymbol{\theta}^{\mathsf{T}}\mathbf{a}^{(\pi(j))} - \boldsymbol{\theta}^{\mathsf{T}}\mathbf{a}^{(\widehat{\pi}(j_1))})}_{\leq 0} + \underbrace{(\boldsymbol{\theta}^{\mathsf{T}}\mathbf{a}^{(\widehat{\pi}(j_1))} - \widehat{\boldsymbol{\theta}}^{\mathsf{T}}\mathbf{a}^{(\widehat{\pi}(j_1))})}_{\leq 2\epsilon}$$

$$+ \underbrace{(\widehat{\boldsymbol{\theta}}^{\mathsf{T}}\mathbf{a}^{(\widehat{\pi}(j_1))} - (\widehat{\boldsymbol{\theta}}^{\mathsf{T}}\mathbf{a}^{(\widehat{\pi}(j))})}_{\leq 0} + \underbrace{(\widehat{\boldsymbol{\theta}}^{\mathsf{T}}\mathbf{a}^{(\widehat{\pi}(j))} - \boldsymbol{\theta}^{\mathsf{T}}\mathbf{a}^{(\widehat{\pi}(j))})}_{\leq 2\epsilon}$$

$$\leq \left\langle \boldsymbol{\theta}^{\mathsf{T}} - \widehat{\boldsymbol{\theta}}^{\mathsf{T}}, \mathbf{a}^{(\pi(j))} \right\rangle + \left\langle \boldsymbol{\theta}^{\mathsf{T}} - \widehat{\boldsymbol{\theta}}^{\mathsf{T}}, \mathbf{a}^{(\pi(j_1))} \right\rangle \leq 4\epsilon$$

Here $\epsilon$ is the error of estimation after exploration. We therefore obtain the following,

$$\mathbb{E}[\text{Reg}(\mathsf{T})] \leq \mathsf{T}_{\text{explore}}R_{\max} + 4(\mathsf{T} - \mathsf{T}_{\text{explore}})\nu\mathsf{O}(\epsilon) + (\mathsf{T} - \mathsf{T}_{\text{explore}})(1 - \nu)R_{\max},$$

Using $\mathsf{T} \gg \mathsf{T}_{\text{explore}}$ (the number of exploration rounds is sublinear in $\mathsf{T}$) we obtain,

$$\mathbb{E}[\text{Reg}(\mathsf{T})] = \mathbb{E}_{\boldsymbol{\theta}}\left[\sum_{t=1}^{\mathsf{T}} \left\langle \boldsymbol{\theta}, \mathbf{a}^{\pi(t)} - \mathbf{a}_t \right\rangle\right] \leq \mathbb{E}_{\boldsymbol{\theta}}\left[2R_{\max}\mathsf{T}_{\text{explore}} + 2R_{\max}\|\boldsymbol{\theta} - \widehat{\boldsymbol{\theta}}\|_2\mathsf{T}\right] \quad (10)$$

**Step 2. Fast Sparse Learning:** We use Theorem 1, which is proved in the appendix, to obtain an estimation guarantee in terms of the number of exploration rounds. And we now apply the bound from Corollary 1 and obtain the following (Making an assumption similar to Hao et al. (2020) on the exploration rounds $\mathsf{T}_{\text{explore}} > \mathsf{O}(k\log^4 M) > \mathsf{O}(k\lambda_{\min}^{-4})$ [3].

**Step 2. Exploration Period Optimization:** ( The probability of error terms $(1 - \nu)$ are left out in the expression.) equation 5 can then be bounded as,

$$\mathbb{E}[\text{Reg}(\mathsf{T})] = \widetilde{\mathsf{O}}\left(\mathsf{T}_{\text{explore}} + \mathsf{T}\mathsf{T}_{\text{explore}}^{-1/2}\lambda_{\min}^{-1}k^{1/2} + \mathsf{T}_{\text{explore}}^{-1/4}\beta^{1/2}\lambda_{\min}^{-1/2} + k^{-1/2}\beta\right)$$

Setting $\mathsf{T}_{\text{explore}} = \widetilde{\mathsf{O}}(k^{1/3}\mathsf{T}^{2/3})$ we obtain the desired result.

$\square$

We can also obtain a regret bound for the case of hard sparsity which is of the same order as Hao et al. (2020).

**Corollary 2.** *Let $\boldsymbol{\theta}$ be $k$-sparse, $\|\boldsymbol{\theta}\|_0 \leq k$ in the sparse linear bandits framework of Theorem 2. Let $\lambda_{\min}^*$ be the minimum eigenvalue from equation 2 with the same assumptions as Theorem 2. Then Algorithm* BSLB *with exploration period $\mathsf{T}_{explore} = \mathsf{O}(k^{\frac{1}{3}}\mathsf{T}^{\frac{2}{3}})$, achieves a regret guarantee of $\mathbb{E}[\text{Reg}(\mathsf{T})] = \mathsf{O}((\lambda_{\min}^*)^{-1}k^{\frac{1}{3}}\mathsf{T}^{\frac{2}{3}})$.*

### A.1.5 PROOF OF THEOREM 3

For vectors $\mathbf{v}$ and $\mathbf{z}$, define $Z_{\mathbf{v}}(\mathbf{z}) = \mathbf{z}^{\mathsf{T}}\mathbf{v}\mathbf{v}^{\mathsf{T}}\mathbf{z}$.

Then the minimum eigenvalue for (covariance matrix of) a set of vectors $\mathcal{G}$ is given by, $\lambda_{\min}(\mathcal{G}) = \min_{\mathbf{z}\in\mathcal{B}^d} \frac{1}{|\mathcal{G}|}\sum_{\mathbf{v}\in\mathcal{G}} Z_{\mathbf{v}}(\mathbf{z})$.

Let randomized rounding be run with $\hat{g}$, and $\hat{\mathcal{G}}$ be the sampled set of arms. Of-course $|\hat{\mathcal{G}}|$ need to be equal to $\hat{g}$. However, we first assume that the denominator of $\lambda_{\min}(\hat{\mathcal{G}})$ is equal to $\hat{g}$. We later show that this assumption worsens the approximation guarantees by 2 with high probability. Under this

---

[3]Another way to analyze this would be to assume $\mathsf{T}_{\text{explore}} = \mathsf{O}(k\mathsf{T}^{2/3})$ which would have similar regret guarantees with a slightly better dependence on $k$ ($k^{-1/12}$ term becomes $k^{-1/4}$) but the results still holds.

assumption by construction of the randomized rounding procedure, the expected minimum eigenvalue of the sampled set is equal to the minimum eigenvalue corresponding to the solution of the convex optimization problem, since, $\mathbb{E}\frac{1}{\hat{g}}Z_{\mathbf{v}}(\mathbf{z}) = \boldsymbol{\mu}_{\mathbf{v}}Z_{\mathbf{v}}(\mathbf{z})$.

We therefore prove the following result to bound the approximation error between the $\lambda_{\min}(\hat{\mathcal{G}})$ obtained from the randomized rounding solution and the optimal solution of the convex relaxation from equation 6.

**Lemma 4.** *Let $\mathcal{A}$ be a set of* $\mathsf{M}$ *arms where each arm is* $\mathbf{a} \in \mathcal{B}^d$ *and let* $Z_{\mathbf{v}}(\mathbf{z}) = \mathbf{z}^\top \mathbf{v}\mathbf{v}^\top \mathbf{z}$. *Let* $\boldsymbol{\mu}$ *be the solution of the convex relaxation of equation 6 at* $\hat{g}$ *and* $\hat{\mathcal{G}}$ *be the set sampled using randomized rounding (Step 18-20 in Alg. 1). Then the following holds,*

$$\mathbb{P}\left(\left|\inf_{z\in\mathcal{B}^d}\frac{1}{\hat{g}}\sum_{\mathbf{v}\in\hat{\mathcal{G}}}Z_{\mathbf{v}}(\mathbf{z}) - \inf_{z\in\mathcal{B}^d}\sum_{\mathbf{v}\in\mathcal{A}}\boldsymbol{\mu}_{\mathbf{v}}Z_{\mathbf{v}}(\mathbf{z})\right| \geq \mathsf{O}\left(\frac{\sqrt{d}\log\mathsf{M}}{\sqrt{|\hat{g}|}}\right)\right) \leq \frac{1}{\log\mathsf{M}} \qquad (11)$$

*Proof.* Using $\mathbb{E}\frac{1}{\hat{g}}Z_{\mathbf{v}}(\mathbf{z}) = \boldsymbol{\mu}_{\mathbf{v}}Z_{\mathbf{v}}(\mathbf{z})$ from the preceding paragraph.

By symmetrization,

$$\mathbb{E}\left[\sup_{z\in\mathcal{B}^d}\left|\sum Z_{\mathbf{v}}(\mathbf{z}) - \mathbb{E}\sum Z_{\mathbf{v}}(\mathbf{z})\right|\right] \leq 2\mathbb{E}\left[\sup_{z\in\mathcal{B}^d}\left|\sum \xi_{\mathbf{v}}Z_{\mathbf{v}}(\mathbf{z})\right|\right]$$

where $\xi_{\mathbf{v}}$ are i.i.d. Rademacher random variables (we overload $\mathbf{v}$ as the index). Now using Dudley's integral inequality,

$$\mathbb{E}\left[\sup_{z\in\mathcal{B}^d}\left|\sum \xi_{\mathbf{v}}Z_{\mathbf{v}}(\mathbf{z})\right|\right] \leq \zeta_{11}\Psi\log^{1/2}(\frac{3}{\epsilon})\sqrt{d}$$

where $\Psi$ is the constant which satisfies,

$$\left\|\sum \xi_{\mathbf{v}}Z_{\mathbf{v}}(\mathbf{z}_1) - \sum \xi_{\mathbf{v}}Z_{\mathbf{v}}(\mathbf{z}_2)\right\|_\psi \leq \Psi\|z_1 - z_2\|_2 \leq \sqrt{2}\Psi$$

Now w.l.o.g.,

$$\left\|\sum \xi_{\mathbf{v}}Z_{\mathbf{v}}(\mathbf{z}_1) - \sum \xi_{\mathbf{v}}Z_{\mathbf{v}}(\mathbf{z}_2)\right\|_\psi \leq 2\left\|\sum \xi_{\mathbf{v}}Z_{\mathbf{v}}(\mathbf{z}_1)\right\|_\psi$$

Now from the definition of subgaussian norm,

$$\left\|\sum \xi_{\mathbf{v}}Z_{\mathbf{v}}(\mathbf{z}_1)\right\|_\psi = \inf\{t : \exp\frac{(\sum \xi_{\mathbf{v}}Z_{\mathbf{v}}(\mathbf{z}_1))^2}{t^2} \leq 2\}$$

Now,

$$\exp\frac{(\sum \xi_{\mathbf{v}}Z_{\mathbf{v}}(\mathbf{z}_1))^2}{t^2} \leq \exp\frac{(\sum Z_{\mathbf{v}}(\mathbf{z}_1))^2}{t^2} \leq \exp 2\frac{\sum(Z_{\mathbf{v}}(\mathbf{z}_1))^2}{t^2} = \prod\exp 2\frac{(Z_{\mathbf{v}}(\mathbf{z}_1))^2}{t^2}\text{(independence)}$$

Now by Chernoff's bound,

$$\prod\exp 2\frac{(Z_{\mathbf{v}}(\mathbf{z}_1))^2}{t^2} \leq \prod\exp\left\{|\mathcal{G}|\boldsymbol{\mu}_{\mathbf{v}}\left(\exp\left(2\frac{(\mathbf{z}^\top\mathbf{v}\mathbf{v}^\top\mathbf{z})^2}{|\mathcal{G}|^2 t^2}\right) - 1\right)\right\} = \exp\left\{|\mathcal{G}|\left(\exp\left(2\frac{(\mathbf{z}^\top\mathbf{v}\mathbf{v}^\top\mathbf{z})^2}{|\mathcal{G}|^2 t^2}\right) - 1\right)\right\}$$

To find $\inf\{t : \exp(\frac{\cdot}{t^2}) \leq 2\}$

$$\exp\left\{|\mathcal{G}|\left(\exp\left(2\frac{(\mathbf{z}^\top\mathbf{v}\mathbf{v}^\top\mathbf{z})^2}{|\mathcal{G}|^2 t^2}\right) - 1\right)\right\} \leq 2 \implies \exp\left(2\frac{(\mathbf{z}^\top\mathbf{v}\mathbf{v}^\top\mathbf{z})^2}{|\mathcal{G}|^2 t^2}\right) \leq \frac{\ln 2}{|\mathcal{G}|} + 1$$

$$(\mathbf{z}^\top\mathbf{v}\mathbf{v}^\top\mathbf{z})^2\frac{2}{\ln\left(\frac{\ln 2}{|\mathcal{G}|} + 1\right)|\mathcal{G}|^2} \leq t^2 \implies \frac{\sqrt{2}\mathbf{z}^\top\mathbf{v}\mathbf{v}^\top\mathbf{z}}{\sqrt{\ln\left(\frac{\ln 2}{|\mathcal{G}|} + 1\right)|\mathcal{G}|}} \leq t$$

Upper bounding $\mathbf{z}^\top\mathbf{v}\mathbf{v}^\top\mathbf{z} \leq 1$, we obtain the $\inf t$,

$$\frac{\sqrt{2}}{\sqrt{\ln\left(\frac{\ln 2}{|\mathcal{G}|} + 1\right)|\mathcal{G}|}} \leq t$$

with $|\mathcal{G}| >> \ln 2, \ln\left(\frac{\ln 2}{|\mathcal{G}|} + 1\right) \approx \frac{\ln 2}{|\mathcal{G}|}$,

$$\frac{\sqrt{2}}{\sqrt{|\mathcal{G}|}} \leq t$$

Therefore combining,

$$\mathbb{E}\left[\sup_{z \in \mathcal{B}^d}\left|\sum Z_{\mathbf{v}}(\mathbf{z}) - \mathbb{E}\sum Z_{\mathbf{v}}(\mathbf{z})\right|\right] \leq 2\sqrt{2}\zeta_{11}\log^{1/2}(\frac{3}{\epsilon})\frac{d^{1/2}}{\sqrt{|\mathcal{G}|}} = \zeta_{10}\frac{\sqrt{d}}{\sqrt{|\mathcal{G}|}}$$

Using Markov's inequality and the above lemma,

$$\mathbb{P}(\sup_{z \in \mathcal{B}^d}\left|\sum Z_{\mathbf{v}}(\mathbf{z}) - \mathbb{E}\sum Z_{\mathbf{v}}(\mathbf{z})\right| \geq \zeta_{10}\frac{\sqrt{d}\log \mathsf{M}}{\sqrt{|\mathcal{G}|}}) \leq \frac{1}{\log \mathsf{M}}$$

If,

$$\sup_{z \in \mathcal{B}^d}\left|\sum Z_{\mathbf{v}}(\mathbf{z}) - \mathbb{E}\sum Z_{\mathbf{v}}(\mathbf{z})\right| \leq \epsilon$$

is true with probability $1 - \delta$. Then $\forall z \in \mathcal{B}^d$,

$$\mathbb{E}\sum Z_{\mathbf{v}}(\mathbf{z}) - \epsilon \leq \sum Z_{\mathbf{v}}(\mathbf{z})$$

$$\inf_{z \in \mathcal{B}^d}\mathbb{E}\sum Z_{\mathbf{v}}(\mathbf{z}) - \epsilon \leq \inf_{z \in \mathcal{B}^d}\sum Z_{\mathbf{v}}(\mathbf{z})$$

with probability $1 - \delta$. Similar to the other direction, we obtain the desired result. $\square$

Then, we bound the size of the actual sampled set with respect to $\hat{g}$, at which the convex relaxation is computed. We derive the following result which gives a probability bound on the number of arms sampled.

**Lemma 5.** *Let $\hat{g}$ be the subset size that the randomized rounding is run with (Line 20 in Alg. 1) and let $\hat{\mathcal{G}}$ be the true number of sampled arms. Then the following probability holds,*

$$\mathbb{P}(\frac{\hat{g}}{2} \leq |\hat{\mathcal{G}}| \leq 2\hat{g}) \geq 1 - 2(\frac{2}{e})^{\frac{\hat{g}}{2}} \tag{12}$$

*Proof.* We prove the following two tail bounds and then take the union bound over them both,

$$\mathbb{P}(|\hat{\mathcal{G}}| \geq 2\hat{g}) \leq (\frac{e}{3})^{\hat{g}}, \mathbb{P}(|\hat{\mathcal{G}}| \leq \hat{g}/2) \leq (\frac{2}{e})^{\frac{\hat{g}}{2}}$$

First the size of the sampled subset is the sum of independent Bernoulli random variables, $|\hat{\mathcal{G}}| = \sum p_j$ where each $p_j = Ber(\boldsymbol{\mu}_j\hat{g})$. Using tail bound from Chernoff bound,

$$\mathbb{P}(|\hat{\mathcal{G}}| \geq 2\hat{g}) \leq \inf_{t>0}\exp(-t2\hat{g})\mathbb{E}[\exp(t|\hat{\mathcal{G}}|)]$$

$$= \inf_t \exp(-t2\hat{g})\prod_j \mathbb{E}[\exp(tp_j)] \text{ (independent rv)}$$

$$\inf_t \exp(-t2\hat{g})\prod_j \mathbb{E}[\exp(tp_j)] \leq \inf_t \exp(-t2\hat{g})\prod_j \exp(\hat{g}\boldsymbol{\mu}_j(\exp(t) - 1))$$

$$= \inf_t \exp(-t2\hat{g})\exp(\hat{g}(\exp(t) - 1))$$

Achieves infinum for $t = \ln 2$,

$$\mathbb{P}(|\hat{\mathcal{G}}| \geq 2\hat{g}) \leq \exp(-2\hat{g}\ln 2 + \hat{g}) = (\frac{3}{e})^{\hat{g}}$$

Using a similar left tail bound,

$$\mathbb{P}(|\hat{\mathcal{G}}| \leq \hat{g}/2) \leq \inf_{t<0} \exp(-t\hat{g}/2)\mathbb{E}[\exp(t|\hat{\mathcal{G}}|)]$$

$$= \inf_{t} \exp(-t\hat{g}/2) \prod_{j} \mathbb{E}[\exp(tp_j)] \text{ (independent rv)}$$

$$\inf_{t} \exp(-t\hat{g}/2) \prod_{j} \mathbb{E}[\exp(tp_j)] \leq \inf_{t} \exp(-t\hat{g}/2) \prod_{j} \exp(\hat{g}\boldsymbol{\mu}_j(\exp(t)-1))$$

$$= \inf_{t} \exp(-t\hat{g}/2) \exp(\hat{g}(\exp(t)-1))$$

Achieves infinum for $t = -\ln 2$,

$$\mathbb{P}(|\hat{\mathcal{G}}| \leq \hat{g}/2) \leq \exp(\hat{g}(-\frac{1}{2}) + \ln 2\hat{g}/2) = (\frac{2}{e})^{\frac{\hat{g}}{2}}$$

Now for $\hat{g} \geq 1$, $(\frac{2}{e})^{\frac{\hat{g}}{2}} \geq (\frac{3}{e})^{\hat{g}}$, and therefore applying the union bound we obtain the required result.

$$\square$$

Therefore the above lemma helps us prove the following statement,

$$\mathbb{P}\left(\frac{1}{2} \leq \frac{\min_{z\in\mathcal{B}^d} \sum_{\mathbf{v}} \frac{1}{|\hat{\mathcal{G}}|} p_{\mathbf{v}} z^{\mathsf{T}}\mathbf{v}\mathbf{v}^{\mathsf{T}}z}{\min_{z\in\mathcal{B}^d} \sum_{\mathbf{v}\in\mathcal{A}} \frac{1}{\hat{g}} p_{\mathbf{v}} z^{\mathsf{T}}\mathbf{v}\mathbf{v}^{\mathsf{T}}z} \leq 2\right) \geq 1 - 2(\frac{2}{e})^{\frac{\hat{g}}{2}}$$

The above two lemmas help us prove the approximation error of the randomized rounding with respect to a fixed parameter $\hat{g}$. However, the approximation errors need to be with respect to the optimal choice of $\hat{g}$, $g^*$, which is the size of the optimal subset from equation 2.

We claim the following about the solution of the convex relaxation at $\hat{g}$ and $d$,

$$\lambda_{\min}^*(d) \leq \underset{\boldsymbol{\mu}\in\mathcal{P}(\mathcal{A});\|\boldsymbol{\mu}\|_\infty\leq\frac{1}{d}}{\arg\max} \inf_{z\in\mathcal{B}^d} \sum_{\mathbf{v}\in\mathcal{A}} \boldsymbol{\mu}_{\mathbf{v}} Z_{\mathbf{v}}(\mathbf{z}) \leq \underset{\boldsymbol{\mu}\in\mathcal{P}(\mathcal{A});\|\boldsymbol{\mu}\|_\infty\leq\frac{1}{d}}{\arg\max} \inf_{z\in\mathcal{B}^d} \sum_{\mathbf{v}\in\mathcal{A}} \frac{1}{d} Z_{\mathbf{v}}(\mathbf{z})$$

$$\leq \frac{\hat{g}}{d} \underset{\boldsymbol{\mu}\in\mathcal{P}(\mathcal{A});\|\boldsymbol{\mu}\|_\infty\leq\frac{1}{d}}{\arg\max} \inf_{z\in\mathcal{B}^d} \sum_{\mathbf{v}\in\mathcal{A}} \frac{1}{\hat{g}} Z_{\mathbf{v}}(\mathbf{z}) \leq \frac{\hat{g}}{d} \underset{\boldsymbol{\mu}\in\mathcal{P}(\mathcal{A});\|\boldsymbol{\mu}\|_\infty\leq\frac{1}{\hat{g}}}{\arg\max} \inf_{z\in\mathcal{B}^d} \sum_{\mathbf{v}\in\mathcal{A}} \frac{1}{\hat{g}} Z_{\mathbf{v}}(\mathbf{z})$$

$$\leq \frac{\hat{g}}{d} \underset{\boldsymbol{\mu}\in\mathcal{P}(\mathcal{A});\|\boldsymbol{\mu}\|_\infty\leq\frac{1}{\hat{g}}}{\arg\max} \inf_{z\in\mathcal{B}^d} \sum_{\mathbf{v}\in\mathcal{A}} \frac{1}{\hat{g}} Z_{\mathbf{v}}(\mathbf{z}) \leq \frac{\hat{g}}{d} \underset{\boldsymbol{\mu}\in\mathcal{P}(\mathcal{A});\|\boldsymbol{\mu}\|_\infty\leq\frac{1}{\hat{g}}}{\arg\max} \inf_{z\in\mathcal{B}^d} \sum_{\mathbf{v}\in\mathcal{A}} \boldsymbol{\mu}_{\mathbf{v}} Z_{\mathbf{v}}(\mathbf{z}) = \frac{\hat{g}}{d}\lambda_{\min}^*(\hat{g})$$

$$(13)$$

The last inequality follows from the fact that $\frac{1}{\hat{g}}$ lies is in the feasibility set.

Also since the convex relaxation at $g^*$ is more constrained than the one at $d$,

$$\lambda_{\min}^*(g^*) \leq \lambda_{\min}^*(d) \leq \frac{\hat{g}}{d}\lambda_{\min}^*(\hat{g}) \implies \lambda_{\min}^*(\hat{g}) \geq \frac{d}{\hat{g}}\lambda_{\min}^*(g^*)$$

We can now combine the two lemmas and the equation above to say that for an error$\epsilon$ and $\hat{g} = \zeta\frac{d\log^2 \mathsf{M}}{\epsilon^2}$ the following holds with high probability,

$$\frac{\epsilon^2}{\zeta \log^2 \mathsf{M}}\lambda_{\min}^* - \frac{1}{2}\epsilon \leq \hat{\lambda_{\min}}$$

where $\zeta$ is a constant.

Further bounding the lower bound,

$$\frac{\epsilon^2}{\zeta \log^2 \mathsf{M}}\lambda_{\min}^* - \frac{1}{2}\epsilon \geq \frac{\lambda_{\min}^*}{\log^2 \mathsf{M}}$$

$$\zeta_1\epsilon^2 - \frac{1}{2}\epsilon - \zeta\zeta_1 \geq 0$$

which has a positive determinant and can only satisfied when $\epsilon \notin \left[\frac{1/2-\sqrt{1/4+\zeta\zeta_1^2}}{2\zeta_1}, \frac{1/2+\sqrt{1/4+\zeta\zeta_1^2}}{2\zeta_1}\right]$.

Since $\epsilon > 0$, therefore for this to be true, $\epsilon > \frac{1/2+\sqrt{1/4+\zeta\zeta_1^2}}{2\zeta_1}$

$$\epsilon > \frac{1/2+\sqrt{1/4+\zeta(\frac{\lambda_{\min}^*}{\zeta\log^2 M})^2}}{2\frac{\lambda_{\min}^*}{\zeta\log^2 M}} = \frac{\zeta\log^2(M)/2+\sqrt{\zeta^2\log^4 M/4+\zeta(\lambda_{\min}^*)^2}}{2\lambda_{\min}^*}$$

If $\lambda_{\min}^* > \zeta\log^2(M)/2$, then we can assume the stronger condition (note that constant $\zeta$ from Lemma 4 is $> 1$),

$$\epsilon > (1+\sqrt{2\zeta})$$

The search bound, therefore becomes, $\hat{g} \leq \frac{d\log^2 M\zeta}{(1+\zeta^2+2\sqrt{2\zeta})} \leq \frac{d\log^2 \lambda_{\min}}{(1+\zeta^2+2\sqrt{2\zeta})}, \implies \hat{g} = O(d\log^2 M)$.

This completes the proof of the theorem.

However if $\lambda_{\min}^* < \zeta\log^2(M)/2$,

Then $\epsilon > \frac{(1+\sqrt{2\zeta})\zeta\log^2(M)}{4\lambda_{\min}^*}$

and the search bound becomes,

$$\hat{g} \leq \frac{16d(\lambda_{\min}^*)^2}{(1+\zeta^2+2\sqrt{2\zeta})\zeta\log^2(M)} \implies \hat{g} = O(\frac{d(\lambda_{\min}^*)^2}{\log^2 M})$$

Therefore for $\zeta_2 = \min(\lambda_{\min}^*, \log^2 M)$, $\hat{g} = O(\frac{d\zeta_2}{\log^2 M})$.

Another way to derive a tighter bound on $\lambda^*$ is,

We consider the stronger condition,

$$\epsilon > \frac{2\sqrt{\zeta^2\log^4 M/4+\zeta(\lambda_{\min}^*)^2}}{2\lambda_{\min}^*}$$

Plugging this into $\hat{g}$,

$$\hat{g} \leq \frac{d\log^2 M(\lambda_{\min}^*)^2}{(\zeta\log^4 M/4+(\lambda_{\min}^*)^2)}$$

From $\hat{g} \geq d$,

$$\lambda_{\min}^* \geq \sqrt{\frac{\zeta\log^4 M}{\log^2 M-1}}$$

Under a stronger condition,

$$\lambda_{\min}^* \geq \sqrt{\frac{2\zeta\log^4 M}{\log^2 M}} = \sqrt{2\zeta}\log M \implies \lambda_{\min} = O(\log M)$$

### A.1.6    BRUTE FORCE ALGORITHM FOR SEARCHING THE OPTIMAL SUBSET

From the previous previous proof we can set $\hat{g} = \frac{4\zeta d\log^2 M}{(\lambda_{\min}^*)^2}$, and then search for subsets in the range $[d, \hat{g}]$ to obtain a minimum eigenvalue $\hat{\lambda_{\min}}$. We obtain the approximation guarantee, $\frac{1}{2}\lambda_{\min}^* \leq \hat{\lambda_{\min}}$ w.h.p., since we are only using the approximation guarantee from Lemma 4 and Lemma 5, and not from equation 13 because we are already searching the space $< \hat{g}$. Since the search space is dependent on $\mathsf{Poly}(d)$, the time complexity of the brute force algorithm, Algorithm 2 follows. This time complexity is substanially smaller than the complexity over the search over all arms which is of the order $O(\exp(M))$

*What if the maximum minimum eigenvalue $\lambda_{\min}^*$ is not known ?* We can use a lower bound on the $\lambda_{\min}^*$. This is easy to obtain: Randomly sample subsets of the arm set $\mathcal{A}$ and compute the objective

---

**Algorithm 2** Brute Force Search for Optimal Subset

---

*Input* Approximation Factor $\epsilon$, Search Bound $\hat{g}$
*Output* Subset $\bar{\mathcal{G}}$
Set $\bar{\lambda}_{\min} = 0, \bar{\mathcal{G}} = \phi$
**for** $\bar{d}$ in $\{d, \dots, \hat{g}\}$ **do**
    **for** $\mathcal{G}'$ in $\{\mathcal{G} \subseteq \mathcal{A}; |\mathcal{A}| = \bar{d}\}$ **do**
        **if** $\lambda_{\min}(\sum_{\mathbf{a} \in \mathcal{G}'} |\mathcal{G}'^{-1}| \mathbf{a}\mathbf{a}^{\mathsf{T}}) > \bar{\lambda}_{\min}$ **then**
            Set $\lambda_{\min} = \lambda_{\min}(\sum_{\mathbf{a} \in \mathcal{G}'} |\mathcal{G}'^{-1}| \mathbf{a}\mathbf{a}^{\mathsf{T}}), \bar{\mathcal{G}} = \mathcal{G}'$
        **end if**
    **end for**
**end for**

---

value in Equation 2 for each subset - the lower bound can be the maximum objective value across the sampled subsets.

*What can the practitioner do to select a good subset size empirically?* Additionally, if a practitioner wants to test out a particular choice of $\hat{g}$, the worst-case error can be empirically calculated (the difference between the convex relaxation at $d$ and averaged across multiple randomized rounding runs for different values of $\hat{g}$). This is possible because GETGOODSUBSET($\mathcal{A}$) can be run offline.

### A.1.7 WHY DOES THE AVERAGE MINIMUM EIGENVALUE CONSTRAINT NOT SATISFY MATROID CONSTRAINTS

Existing work in experiment design work with objective functions which often satisfy the matroid, submodularity or cardinality constraints to perform experiment design (Allen-Zhu et al., 2021; Brown et al., 2024). However we need to optimize the minimum eigenvalue averaged across the subset (because we want to avoid dependence of M in the regret term and also use the RE condition). Our objective clearly does not satisfy the cardinality constraint, and the feasible sets don't satisfy a matroid constraint (since removing a vector might improve the minimum eigenvalue averaged across the set, so there is no clear partitioning/structure to the feasible set). Finally, we tried to prove submodularity but were unable to do so for our objective function especially because the denominator is dependent on the subset size.

### A.1.8 DETAILS ON CORRALLING

*Brief Overview on Corralling:* The algorithm CORRAL (Agarwal et al., 2017) is a meta-bandit algorithm that uses online mirror descent and importance sampling to sample different bandit algorithms that receive the reward. Using the rewards updates the probabilities used for sampling. The main objective is to achieve a regret which is as good as if the best base algorithm was run on its own.

To setup the context, we exactly reproduce the following excerpt, definition and theorems have been taken from Agarwal et al. (2017):

*For an environment $\mathcal{E}$, we define the environment $\mathcal{E}'$ induced by importance weighting, which is the environment that results when importance weighting is applied to the losses provided by environment $\mathcal{E}$. More precisely, $\mathcal{E}'$ is defined as follows. On each round $t = 1, \dots, T$,*

*1. $\mathcal{E}'$ picks an arbitrary sampling probability $p_t \in (0, 1]$ and obtains $(x_t, f_t) = \mathcal{E}(\theta_1, \dots, \theta_{t-1})$.*

*2. $\mathcal{E}'$ reveals $x_t$ to the learner and the learner makes a decision $\theta_t$.*

*3. With probability $p_t$, define $f'_t(\theta, x) = f_t(\theta, x)/p_t$ and $\theta'_t = \theta_t$; with probability $1 - p_t$, define $f'_t(\theta, x) = 0$ and $\theta'_t \in \Theta$ to be arbitrary.*

*4. $\mathcal{E}'$ reveals the loss $f'_t(\theta_t, x_t)$ to the learner, and passes $\theta'_t$ to $\mathcal{E}$.*

**Definition 3.** *(Agarwal et al., 2017) For some $\alpha \in (0, 1]$ and non-decreasing function $R : \mathbb{N}_+ \to \mathbb{R}_+$, an algorithm with decision space $\mathcal{O}$ is called $(\alpha, R)$-stable with respect to an environment $\mathcal{E}$ if its regret under $\mathcal{E}$ is $R(T)$, and its regret under any environment $\mathcal{E}'$ induced by importance weighting*

*is*

$$\sup_{\theta \in \Theta} \mathbb{E} \left[ \sum_{t=1}^{T} f_t(\theta, x_t) - f_t(\theta, x_t) \right] \le \mathbb{E}[\rho^{\alpha}] R(T) \qquad (2)$$

*where $\rho = \max_{t \in [T]} 1/p_t$ (with $p_t$ as in the definition of $\mathcal{E}'$ above), and all expectations are taken over the randomness of both $\mathcal{E}'$ and the algorithm.*

Similar too most reasonable Base Algorithms it can be seen that the BSLB algorithm satisfies is $(1, \mathbb{E}[\mathsf{Reg}(\mathsf{T})])$-stable by rescaling the losses.

We

**Theorem 8.** *(Theorem 4 in (Agarwal et al., 2017)) For any $i \in [M]$, if base algorithm $\mathcal{B}_i$ (with decision space $\mathcal{O}_i$) is $(\alpha_i, R_i)$-stable (recall Defn. 3) with respect to an environment $\mathcal{E}$, then under the same environment CORRAL satisfies*

$$\sup_{\theta \in \Theta,\ \pi \in \Pi} \mathbb{E} \left[ \sum_{t=1}^{T} f_t(\theta_t, x_t) - f_t(\theta, x_t) \right] = \widetilde{O} \left( \frac{M}{\eta} + T\eta \frac{|\mathcal{O}_{\pi_t}|}{\eta} + \frac{\alpha_i}{\eta\beta} R_i(T) \right), \qquad (3)$$

*where all expectations are taken over the randomness of CORRAL Algorithm, the base algorithms, and the environment.*

**Theorem 9.** *(Theorem 5 in (Agarwal et al., 2017)) Under the conditions of Theorem 7, if $\alpha_i = 1$, then with $\eta = \min\left\{ \frac{1}{4\theta R_i(T) \ln T}, \sqrt{\frac{1}{T}} \right\}$ CORRAL satisfies:*
$\sup_{\theta \in \Theta, \pi \in \Pi} \left[ \sum_{t=1}^{T} f_t(\theta_t, \pi_t) - f_t(\theta, \pi_t) \right] = \widetilde{O} \left( \sqrt{MT} + M R_i(T) \right).$

A.1.9 PROOF OF THEOREM 4

---

**Algorithm 3** Corralling with Blocked Linear Sparse Bandits (`C-BSLB`)

---

1: Input: Dimension $d$, Total Number of Rounds $\mathsf{T}$, Regret Bound of Best Algorithm $R_{\text{best}}$
2: Set Learning rate $\eta = \min \left( \frac{1}{40\mathsf{T} R_{\text{best}}}, \sqrt{\frac{\lfloor \log_2(d) \rfloor}{\mathsf{T}}} \right)$
3: Set Exponential Grid $k \in [1, 2, \ldots 2^{\lfloor \log_2(d) \rfloor}]$
4: Initialize $\lfloor \log_2(d) \rfloor + 1$ Base Algorithms one for each sparsity parameter on an exponential grid, `BSLB`($\mathsf{T}_{\text{explore}} = \zeta k^{1/3} \mathsf{T}^{2/3}$)
5: Sample $\mathsf{M}_{\text{sampled}} = \zeta d^{1/3} \mathsf{T}^{2/3}$ arms without replacement to be used as proxy samples.
6: Run Corral($\lfloor \log_2(d) \rfloor + 1$ `BSLB` algorithms, $\eta$) from Agarwal et al. (2017) with Base Algorithms and time horizon $\mathsf{T}$. If an arm is suggested which is already pulled, pull an arm from the remaining set of arms uniformly at random.

---

Before presenting the proof, we clarify what we mean by the exponential scale with an example. For dimension $d = 1024$, the exponential scale will be $k \in \{1, 2, 4, 8, 16, 32, 64, 128, 256, 512, 1024\}$, and we initialize a base algorithm each with the exploration period set according to the $k$.

*Remark:* Also Step 5 and Step 6 in Algorithm 3 needs explanation. Note that the CORRAL algorithm as a whole has to respect the *blocking* constraint. Even though CORRAL does not require the bandit algorithms to be run independently, we want to avoid changing CORRAL or the base algorithm. Instead we just change the arms that are available to sample by exploiting the fact that our base algorithm is a two step algorithm and each of the steps can be performed offline. For each base algorithm:

1. For the explore phase of the base algorithm we take arms from the intersection of the subset sampled in step 5 and the subset of arms which have not been sampled.

2. For the exploit phase if the chosen base algorithm provides an arm which has already been sampled by CORRAL. Then we provide the feedback corresponding to that arm. And then we pull an arm from the remaining set of arms without replacement.

Note that with this modification, the exploration phase of each of the algorithm runs as if the algorithm was being run independently. Hence the regret bounds for each individual base algorithm still holds. We can now prove Theorem 9.

We now use Theorem 9 with $M = \log_2\lceil d\rceil + 1$ algorithms. However if we simply apply the theorem we can bound with respect to the sparsity parameter which lies on the grid, $s \in \{2^i\}_{i=0}^{\log_2\lceil d\rceil}$,

$$\mathbb{E}[\mathsf{Reg}(\mathsf{T})] \leq \tilde{\mathsf{O}}(\sqrt{\log_2\lceil d\rceil\mathsf{T}} + \log_2\lceil d\rceil\mathbb{E}[\mathsf{Reg}(\mathsf{T})]_s)$$

But the optimal sparsity parameter $k^\star$ may not lie on the grid and we need to bound $\mathbb{E}[\mathsf{Reg}(\mathsf{T})]_{k^\star}$ in terms of $\mathbb{E}[\mathsf{Reg}(\mathsf{T})]_s$. To that end we prove the following lemma,

**Lemma 6.** *Let $k^\star$ be the sparsity parameter at which the regret bound of 5 is minimized. And let $s \in \{2^i\}_{i=0}^{\log_2\lceil d\rceil}$ be the parameter on grid which is closest to $k^\star$ in absolute distance. Then the following holds (where $\nu$ is the probability of exploration round succeeding at sparsity level $k^\star$),*

$$\mathbb{E}[\mathsf{Reg}(\mathsf{T})]_s \leq \sqrt{2k^*}\mathbb{E}[\mathsf{Reg}(\mathsf{T})]_{k^\star} + \log_2(2k^*)\mathsf{O}(1-\nu)$$

*Proof.* Let the bound on the expected regret for sparsity level $k$ be given by $\mathbb{E}[\mathsf{Reg}(\mathsf{T})]_k$.

From the statement of the theorem, we assume that for the optimal sparsity parameter $k^\star$, the nearest parameter (on the exponential scale) is $s$. $k^\star$ lies in the interval $[\lceil s/2\rceil, 2s]$ (otherwise $s$ would not be the closest parameter on the exponential scale.). Therefore if we were perform a binary search for $k^\star$, we would need at most $Y = \lfloor\log_2(4k^\star)\rfloor$ queries to search for $k^\star$. Let $k_1^*, k_2^*, \ldots, k_Y^*$ be the mid-points of these queries, where $k_Y^* = k^\star$. Now each of them is such that $k_j^* = \alpha k_{j-1}^*$, where $\alpha \in [0.75, 1.25]$.

First consider the case when $\alpha \in [0.75, 1]$, then by substituting $k = \lfloor\alpha k\rfloor$, in the regret bound of Theorem 2, the following inequalities can be obtained ,

$$\mathbb{E}[\mathsf{Reg}(\mathsf{T})]_{\lfloor\alpha k\rfloor} \leq (\frac{1}{\alpha})^{1/2}\mathbb{E}[\mathsf{Reg}(\mathsf{T})]_k + (1-\nu)\mathsf{O}(\mathsf{T}) \leq \sqrt{2}\mathbb{E}[\mathsf{Reg}(\mathsf{T})]_k + (1-\nu)\mathsf{O}(\mathsf{T}).$$

Now for the case, $\alpha \in [1, 1.25]$, we substitute for $k = \lceil\alpha k\rceil$

$$\mathbb{E}[\mathsf{Reg}(\mathsf{T})]_{\lceil\alpha k\rceil} \leq (\alpha)^{1/3}\mathbb{E}[\mathsf{Reg}(\mathsf{T})]_k + (1-\nu)\mathsf{O}(\mathsf{T}) \leq \sqrt{2}\mathbb{E}[\mathsf{Reg}(\mathsf{T})]_k + (1-\nu)\mathsf{O}(\mathsf{T})$$

The probability of success of each of them is $1 - o(1)$ and $\log(4k^*)$ times the probability of error is still $o(1)$.

Now we can take a cascade of products by decomposing $\mathbb{E}[\mathsf{Reg}(\mathsf{T})]_{k^*}$ using the above inequality in the direction of $k_1^*, k_2^*, \ldots, k_Y^*$. (i.e. we can decompose $k^\star = \alpha_1\alpha_2\ldots\alpha_Y k$,

$$\mathbb{E}[\mathsf{Reg}(\mathsf{T})]_s \leq \alpha\mathbb{E}[\mathsf{Reg}(\mathsf{T})]_{k_1^*} + O(1-\nu) \leq \cdots \leq (\sqrt{2})^Y\mathbb{E}[\mathsf{Reg}(\mathsf{T})]_{k^\star} + Y\mathsf{O}(1-\nu)$$

$$\mathbb{E}[\mathsf{Reg}(\mathsf{T})]_s \leq 2\sqrt{k^\star}\mathbb{E}[\mathsf{Reg}(\mathsf{T})]_{k^\star} + \log_2(4k^\star)\mathsf{O}(1-\nu)$$

$\square$

## A.2 ADDITIONAL NUMERICAL EXPERIMENTS

### A.2.1 PERSONALIZED RECOMMENDATION WITH SINGLE RATING PER ITEM

Next we demonstrate our bandit algorithm on real-world recommendation datasets. However, we construct the recommendation task such that a) each item receives only a single rating from a user and b) we can only use previous recommendations of a user for recommending content. This is in contrast to the standard collaborative filtering setting where an item can where the ratings of the other users is used to recommend content to you. Our setting makes this possible by exploiting the additional information from the embeddings obtained from a pre-trained network for the text (or image) features of the different items. We argue that our setting is be more relevant in recommendation scenarios where privacy is a concern.

We run the corralling algorithm using copies $= 4$ copies of Algorithm 1 each with different exploration periods, $\mathsf{T}_{\text{explore}}$. Each of the instance, we first give $\mathsf{T}_{\text{explore}}$ random recommendations by sampling uniformly without replacement from a suitably constructed subset $\mathcal{G}$ to each user. Given the ratings obtained, we estimate the parameter $\theta_{\text{user}}$ specific to the user using only their recommendations. For the remaining $\mathsf{T}_{\text{exploit}} = \mathsf{T} - \mathsf{T}_{\text{explore}}$ rounds, we give the top $\mathsf{T}_{\text{exploit}}$ recommendations based on the estimated parameter. . To benchmark we run the algorithms independently and also against a random policy which randomly recommends. We next describe the three tasks that we report our results on,

**Goodbooks-10k (Book Reviews):** We use the Goodbooks-10k for a personalized book recommendation tasks (Zajac, 2017). For each book we use the title and the author to obtain embeddings using the MiniLM-L6-v2 sentence transformer which we use as the feature vectors for the arms. There are $\mathsf{M} = 1500$ books and we consider 10 users which have more than $600$ ratings. The ratings are between 1 to 5. We consider the exploration periods as $[100, 150, 200, 300]$.

**MovieLens (Movie Ratings):** MovieLens 100K dataset has $100,000$ ratings (5-star) from $1000$ users on $1700$ movies (Harper & Konstan, 2015). We obtain the embeddings of the TMDB descriptions of the movies using the same sentence transformer model. For experimentation, we consider the users which have more than $300$ ratings and average the results across the users, there are $300$ such users and we run the experiment with 100 different seeds for each user. We consider $\mathsf{T}_{\text{explore}}$ for the different instances in the range $[50, 100, 150, 200]$.

**Jester (Joke Ratings):** We use the Jester joke dataset 1 which has ratings on 100 jokes by $24,983$ users (Goldberg et al., 2001). We obtain embedding for the jokes using the same transformer as above. For experimental purposes we filter out users which do not have ratings on all the jokes and are left with 7200 users. We run our algorithm with 10 different random seeds for each of the 7200 users and report the results averaged across all users. The joke ratings range from $-10$ to $10$. For different algorithm instances $\mathsf{T}_{\text{explore}}$ is taken to be $[20, 40, 60, 80]$ **Results:** We summarize

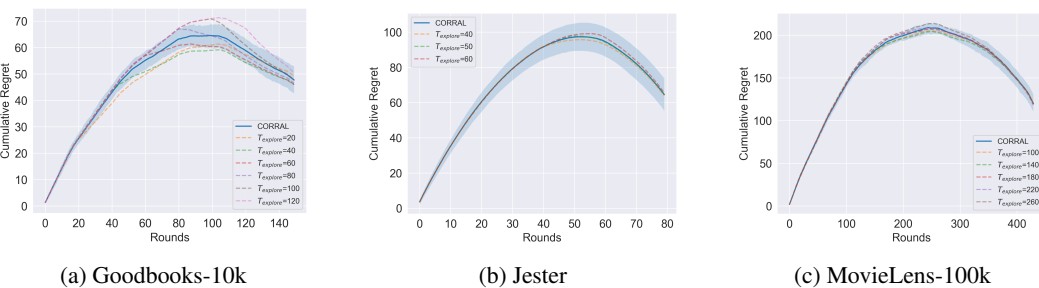

|   |   |   |
|---|---|---|
| (a) Goodbooks-10k | (b) Jester | (c) MovieLens-100k |

Figure 1: Cumulative Regret for recommendation using only single ratings using `BSLB` with different exploration periods and when run with CORRAL (Agarwal et al., 2017). Since the regret is with respect to ranking the arms, it can decrease (since there can be negative terms).

the cummulative regret from Eq. 1 of the algorithms in Figure 1. We add the random policy as a reference. We see that for the different dataset our the algorithm achieves a sub-linear regret. The reason the cumulative regret is not monotonic is due to the fact that the regret is with respect to the top-$\mathsf{T}$ arms. It can be seen that our algorithm with Corral achieves a performance close to the performance of the algorithm with the exploration period, $\mathsf{T}_{\text{explore}}$ out of the $5$.

### A.2.2 HYPERPARAMETERS FOR EXPERIMENT OF SECTION 3

The hyperparameters are presented in Table 2. Note that the reason for selecting different $\mathsf{T}$ across different objects was because the validation datasets had to be big enough (so that the variance of accuracy is informative) and different objects had different total number of samples. The number of exploration rounds are set with respect to $\mathsf{T}(\sim 0.5\mathsf{T} - 0.7\mathsf{T})$ so that the approximation error after exploration is small enough. The active learning methods are also run with random initialization of $\mathsf{T}_{\text{explore}}$ explorations and one round of querying with $\mathsf{T} - \mathsf{T}_{\text{explore}}$ queries. We present the results of a study where we vary only the $\mathsf{T}_{\text{explore}}$ for the same value of $\mathsf{T}$ in Table 3 and observe that with decreasing $\mathsf{T}_{\text{explore}}$ the estimation of difficulty scores deteriorates, and the performance on hard-valid deteriorates. The performance on easy-valid improves since samples are randomly chosen if

the estimation is unsuccessful. Note that our method still performs better than AL baselines and random sampling.

| Object Being Annotated | $N_{\text{valid}}$ (hard) | $N_{\text{valid}}$ (easy) | T | $T_{\text{explore}}$ | All | | BSLB |
|---|---|---|---|---|---|---|---|
| | | | | | Num Samples | Averaged Accuracies | Averaged Accuracies |
| chair | 60 | 80 | 100 | 80 | 960 (10x) | 83.65 | **83.95** |
| car | 70 | 100 | 90 | 60 | 1227 (13x) | 82.05 | **85.25** |
| bottle | 70 | 100 | 120 | 60 | 822 (6x) | 80.8 | **80.8** |
| bottle or chair | 120 | 120 | 140 | 100 | 1807 (13x) | **83.45** | 82.35 |

Table 2: Different hyperparameters used for the experiment of Sec~3. The num samples show how our method achieves a similar accuracy ($-1\%$ to $4\%$ improvement over all) by considering substantially less samples.

| | Validation Type | Object Annotated | AnchorAL | SEALS | Random | All | **Our** (BSLB) |
|---|---|---|---|---|---|---|---|
| $T_{\text{explore}} = 80$ | easy-valid | chair | 94.5±1.0 | 93.2±1.5 | 95.7±1.0 | **96.0±1.4** | 92.3±1.4 |
| | | car | 95.8±2.5 | 95.3±2.0 | 97.3±1.2 | **98.2±1.1** | 95.7±1.7 |
| | | bottle | 95.0±0.9 | 95.0±1.1 | 96.2±1.9 | 96.7±1.8 | **96.8±1.2** |
| | **hard-valid** | chair | 72.0±3.0 | 71.0±1.4 | 67.0±5.0 | 69.4±4.1 | **73.4±2.4** |
| | | car | 66.4±7.9 | 69.2±5.6 | 52.6±8.2 | 58.8±6.8 | **74.0±2.7** |
| | | bottle | 61.2±1.5 | 61.8±2.0 | 51.2±2.9 | 52.6±3.1 | **63.2±2.9** |
| $T_{\text{explore}} = 60$ | easy-valid | chair | 94.8±2.1 | 93.7±1.7 | 95.5±2.0 | **95.5±1.4** | 94.8±1.8 |
| | | car | 96.2±1.9 | 95.5±1.9 | 97.5±1.2 | **98.3±1.1** | 97.0±2.3 |
| | | bottle | 95.0±1.2 | 94.8±1.0 | 97.2±1.7 | **97.5±1.7** | 96.0±1.9 |
| | **hard-valid** | chair | 70.0±5.5 | 70.0±3.8 | 68.0±4.2 | 69.8±4.0 | **72.4±1.7** |
| | | car | 65.8±8.7 | 70.0±7.5 | 53.2±5.4 | 60.6±8.1 | **72.6±3.9** |
| | | bottle | 61.6±1.0 | 61.6±2.3 | 53.4±2.6 | 54.0±2.6 | **62.6±2.8** |
| $T_{\text{explore}} = 30$ | easy-valid | chair | 94.8±1.5 | 94.5±1.2 | 96.3±1.4 | **96.7±1.9** | 94.5±3.2 |
| | | car | 95.3±2.3 | 95.8±1.9 | 97.3±1.2 | **98.2±1.1** | 92.0±11.6 |
| | | bottle | 95.3±0.8 | 95.0±0.5 | 96.2±1.9 | 96.7±1.8 | **96.7±1.2** |
| | **hard-valid** | chair | 69.4±2.1 | **71.2±2.4** | 67.6±4.9 | 69.8±4.0 | 70.8±4.5 |
| | | car | 64.2±9.5 | 67.6±7.7 | 52.6±8.2 | 58.8±6.8 | **70.8±6.5** |
| | | bottle | 60.2±1.9 | 62.0±2.1 | 51.2±2.9 | 52.6±3.1 | **63.0±4.9** |

Table 3: Learning Accuracies on Different Methods for Image Classification in PASCAL-VOC 2012: Effect of $T_{\text{explore}}$ with the number of rounds fixed at $T = 120$ and with 120 **easy-valid** and 100 **hard-valid** samples.

### A.2.3 TEXT CLASSIFICATION ON SST-2

Next we present a result on text classification task on SST-2 (Socher et al., 2013). However since there were no human difficulty ratings available for this task, we use rarity of text (Zhang et al., 2018) as a heuristic for the difficulty ratings. The learning model is a SVM which classifies sentence embeddings obtained from the MiniLM-L6-v2 transformer. We consider $T_{\text{explore}} = 100$ samples and $T_{\text{exploit}} = 200$. The normalized rarity ranges from 0 to 1 and we set $\tau_{\text{hard}} = 0.5$ and $\tau_{\text{easy}} = 0.2$.

We observe a similar trend as the previous task where **BSLB** method performs better than a **random** subset by $3\%$ and as good as the **random-large** subset on the **hard-valid** dataset. There is no regression on the **easy-valid**. The results on both the validation sets are comparable with **mixed** dataset which require all the difficulty ratings (which can be computed for the heuristic but not otherwise). **BSLB** performs better than both active learning methods on both the validation sets by $2\%$. However, since this is a standard sentiment analysis task, the embeddings are more informative, thereby improving the baseline performance for a **random** subset.

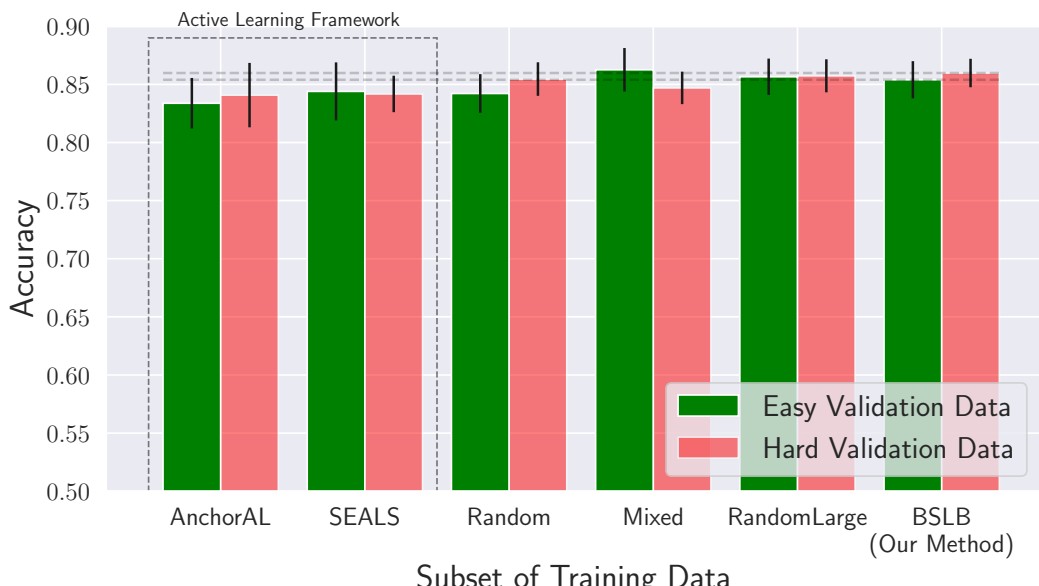

Figure 2: Text Classification on SST-2: The gains are not substantial on the text-classification, but show that our methods are task agnostic. Although conceptually active learning also does adaptive annotation, our method performs better (especially on **hard-valid**) in the label-scarce setting when $\mathsf{T} \ll d$ and the hardness of the samples considered.

### A.2.4  SIMULATION STUDY

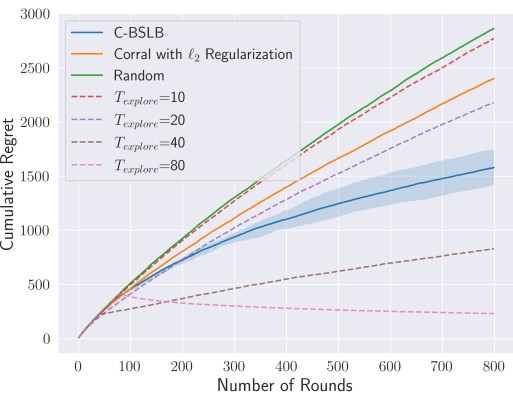

Figure 3: Regret of different algorithms in a Simulated Blocked Sparse Linear Bandit Setup.

Finally, we also run a simulation study to study the efficacy of our `BSLB` and `C-BSLB` algorithm and demonstrate how CORRAL can be used to achieve a sub-linear regret without the knowledge of the optimal parameters. We compute the cumulative regret at time $t$ compared to the top$-t$ arms, and unlike the standard bandit setting, in a blocked setting, the cumulative regret need not be monotonic. To highlight how our method exploits the sparsity of the parameter, we also run CORRAL with multiple versions of our algorithm but with a simple linear regression estimator. We simulate the experimental set-up with the following parameters $\mathsf{M} = 10000, d = 1000$ and $\mathsf{T} = 300$. At sparsity level $k = 10$, the tail parameter is $\gamma = 3$. The experiment is repeated with 100 different random parameter initialization. We plot the cumulative regret in Fig. 3, for algorithms run with different exploration period and two versions of the CORRAL algorithm. Our C-BSLB performs better than corralling with $\ell_2-$regularization, showing that our method exploits the sparsity and does not require true knowledge of the hyperparameters. We also benchmark against a random policy and show that our method performs significantly better showing that the upper bound on regret is not *vacuous*.

### A.2.5 CORRELATION BETWEEN MODEL DIFFICULTY AND HUMAN ANNOTATION DIFFICULTY

In Figure 4a, we show that for the chair images, if a model $\mathcal{M}$ (from the numerical study of Section 3) is trained on all images, then the fraction of difficult samples at a certain distance from the classifier boundary goes down as the distance from the classifier body increases; especially after a certain distance from the decision boundary.

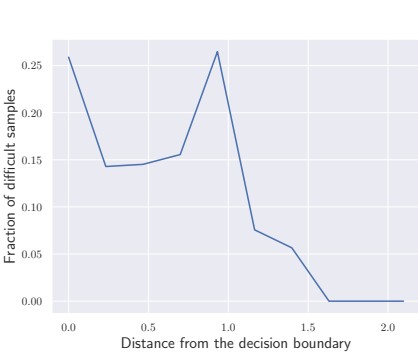

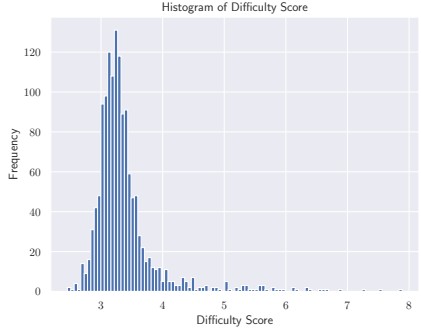

(a) Fraction of difficult samples (labelled by humans) against the distance from decision boundary for SVM trained on all *chair* images. As the distance from the decision boundary increases the fraction of difficult samples (difficulty rating from humans > 3.5) decays to 0.

(b) Histogram shows the heavy-tailed distribution of the difficulty score from Ionescu et al. (2016) of the *chair* object of the PASCAL-VOC dataset. We clip the entries from the middle since they make the difficult estimation noisier, in practical implementation, one would need to develop a mechanism to flag samples with *ambiguous* difficulty and this is left for future work.

### A.2.6 HOW DOES THE TAIL OF THE PARAMETER MATTER?

In Figure 5, we investigate the effect of the tail parameter in the performance of `BSLB` with a fixed exploration period $\mathsf{T}_{\text{explore}} = 80$ and different sizes of the tail in the same setup as the simulation study of Appendix A.2.4. We observe that as the tail parameter $\gamma$ grows, the regret worsens, however we remark that even for a decent $\gamma = 75$, the performance is reasonable.

### A.2.7 CONVERGENCE OF CORRAL PARAMETERS IN C-BSLB

We plot the convergence of the CORRAL parameters of `C-BSLB` in Figure 6 for the simulated experiment of Appendix A.1.8. We observe that the probability of sampling the best algorithm ($\mathsf{T}_{\text{explore}} = T_3 = 80$) increases with rounds. Note that since the experiments were run on a limited resource machine, we could only do $d = 1000$, and for our setup $\mathsf{T} \ll d$ has to be sufficiently low (500 in this case). This is not enough for the CORRAL algorithm to truely exploit the best possible algorithm in `C-BSLB` but as we see in Figure 3, however it still achieves the a decent performance.

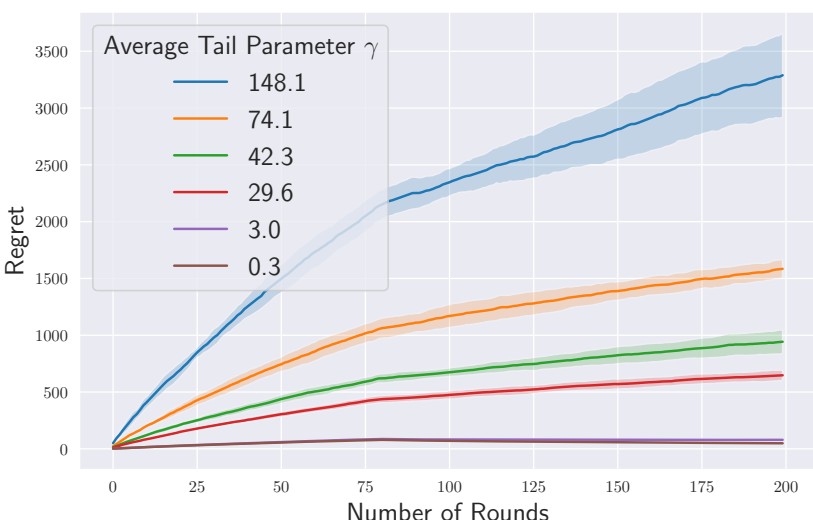

Figure 5: Effect of the tail parameter $\gamma$ on the performance of the BSLB algorithm with $\mathsf{T}_{\text{explore}} = 80$. As the tail increases in magnitude the cummulative regret worsens (increases). However observe that our algorithm is still robust to reasonably large tail $\gamma = 75$.

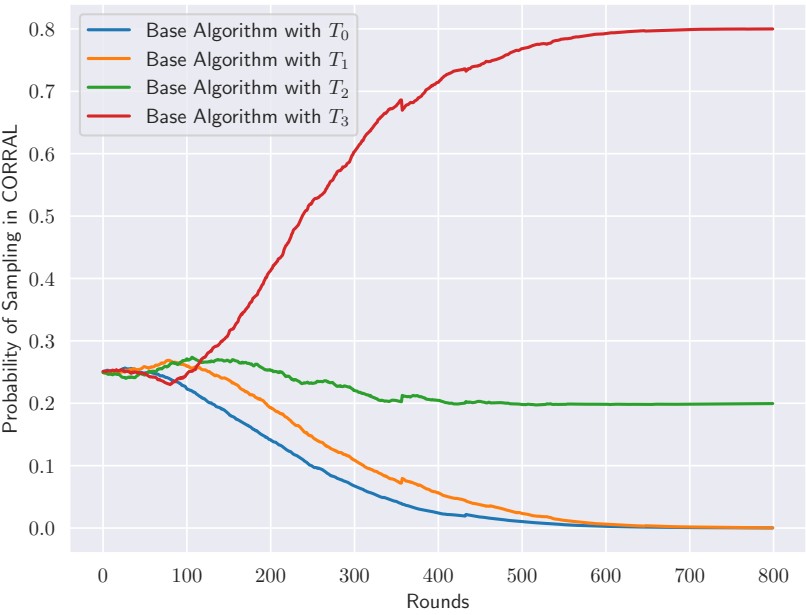

Figure 6: Convergence of the different sampling probabilities for the base algorithms of the C-BSLB (Algorithm 3). This plot is with respect to the simulation study parameters. We can observe that the probability for the best algorithm ($\mathsf{T}_3$) improves with each iteration and for the worst performing algorithm ($\mathsf{T}_0$) decays to 0.

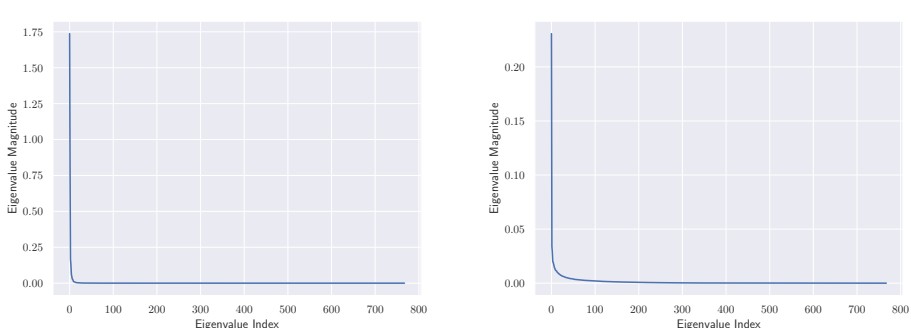

(a) For the PASCAL-2012 on object **chair** with ViT Base Patch16-224 embeddings

(b) Balanced Sample (2500 datapoints) of SST-2 with All MPNet Base V2 embeddings

Figure 7: Eigenvalue spectrum of the embeddings of the two dataset show exponential decay in the eigenvalues, which implies that a uniformly random sample covers the set optimally with high probability because the data is primarily shaped by a few directions.

