# OpenReview forum: "Annotation Efficiency: Identifying Hard Samples via Blocked Sparse Linear Bandits"
_ICLR.cc/2025/Conference — Submitted to ICLR 2025_

### Official Review · Reviewer_9L3c · 2024-11-03

**Soundness:** 3
**Presentation:** 4
**Contribution:** 4
**Rating:** 6
**Confidence:** 2

**Summary:**

This paper studies the sample selection problem in active learning when the label budget is limited. The main contribution is to model this problem as a sparse linear bandit with a blocking constraint. To address this challenge, the authors propose an explore-then-commit algorithm incorporating several novel ingredients. Theoretical analysis demonstrates that the proposed algorithm achieves an $O(k^{1/3} T^{2/3} + k^{-\frac{1}{12}} \beta_k^{1/2} T^{5/6})$ bound and experiments are conducted to validate the effectiveness of the proposed method.

**Strengths:**

- The paper presents an interesting formulation of the sample selection problem in active learning as a sparse linear bandit problem. Several innovative techniques are introduced to derive an algorithm with theoretical guarantees.
- This paper is well-written, with the authors clearly explaining the motivation, technical challenges, and main contributions.
- Empirical studies are conducted to validate the theoretically oriented methods.

**Weaknesses:**

Overall, I do not see any major weaknesses in this paper, though several points are worth discussing:

- **Tightness of the Bound**: As mentioned in lines 349-351, the proposed method achieves the same $T^{2/3}$ regret bound as previous work under the hard sparsity condition. However, the lower bound for the soft sparsity condition remains unclear, and it is uncertain whether the $T^{5/6}$ dependence is tight.
- **Model Assumptions**: The paper considers a scenario where the hardness of the sample is generated from a linear model, which may not always hold in practical settings.

=====post-rebuttal=====
I have reviewed the author's rebuttal and the other reviews. I think the paper introduces several interesting algorithmic components to address the blocking constraint. However, I agree with the other reviewers' concerns about the problem setting and the discrepancy between the paper's goal (training an effective classifier) and its actual focus (identifying hard examples). I have lowered my score to reflect this.

**Questions:**

- Could you provide more discussion on the lower bound of the problem? While establishing a precise lower bound may be challenging, it would be helpful to explain why achieving better results is difficult.
- Could you discuss how to handle cases with model misspecification, where the hardness of the sample is not generated by a linear model?

---

> ### Author Response · Authors · 2024-11-16
> **Response to Review**
>
> We thank the reviewer for acknowledging our  theoretical contributions and for the constructive feedback. Below, we provide clarifications for the questions raised by the reviewer:
>
>
> **Tightness of the Bound: As mentioned in lines 349-351, the proposed method achieves the same regret bound as previous work under the hard sparsity condition. However, the lower bound for the soft sparsity condition remains unclear, and it is uncertain whether the dependence is tight.**
>
> We have now been able to prove a tight $\Omega(\min(k^{1/3}\mathsf{T}^{2/3},\sqrt{d\mathsf{T}}))$ lower bound on the regret for our problem by a reduction of the unblocked setting (studied in Hao et al.) to the blocked problem and subsequently invoking the lower bound in Hao et al. - this lower bound matches the regret upper bound achieved for hard sparse vectors in the blocked setting (see the result in Theorem 2 in paper with tail $\beta_k=0$).  To see the detailed proof, please take a look at Appendix A.1.1 in the updated paper.  However, a lower bound for soft sparse parameter vectors (where the magnitude of the tail is unknown) is challenging and still an open problem.
>
>
> **Model Assumptions: The paper considers a scenario where the hardness of the sample is generated from a linear model, which may not always hold in practical settings.**
>
> The reviewer is absolutely correct. To begin a theoretical study, we have started with the linear setting. This is the simplest of the assumptions that we could make to keep the analysis tractable while having meaningful results. Model misspecification is indeed an interesting avenue for future work.
>
>
> **Could you discuss how to handle cases with model misspecification where the hardness of the sample is not generated by a linear model?**
>
> Following up on the previous response, we can model the hardness score by a more complex model class (say Generalized Linear Models) with relevant structural constraints (analogous to sparsity). The main challenge is to first obtain offline statistical parameter estimation guarantees of an estimator for such a structured model class with few data points under weak assumptions such as RE - such results are very much open for complex model classes.
>
>
> **Could you provide more discussion on the lower bound of the problem? While establishing a precise lower bound may be challenging, it would be helpful to explain why achieving better results is difficult.**
>
> The main difficulty in proving a lower bound for soft sparsity (that goes beyond the vanilla $\Omega(\mathsf{T}^{2/3})$ lower bound proved in this paper update) is to come up with the pair of hard instances (or packing of hard instances) for which parameter distance is large but Total Variation distance is small - even in Hao et al., construction of the hard pair of instances to prove the regret lower bound for hard sparsity without blocking constraints is highly non-trivial. We conjecture that to prove a tight lower bound for soft sparsity, it is necessary to use Fano's inequality in some form for which a careful packing of hard instances in some volume is necessary.

---

> ### Author Response · Authors · 2024-11-22
>
> We thank the reviewer for their objective evaluation of the paper and for appreciating our paper. The reviewer raised some good points, and we have tried answering them in our rebuttal. If the reviewer has further questions, we'd love to address them and provide any clarifications. We again thank the reviewer for their valuable feedback!

---

### Official Review · Reviewer_MzN1 · 2024-11-03

**Soundness:** 3
**Presentation:** 2
**Contribution:** 3
**Rating:** 6
**Confidence:** 2

**Summary:**

This paper considers the problem of minimizing the regret between the hardness of the $T$ selected points and the top $T$ data points that have the top hardness, where $T$ is the budget number of rounds for the human experts to label. The paper treats each data point in the dataset as an arm (using linear bandit) and assumes a blocking constraint that each arm can only be pulled at most once. When the human expert is asked to label the data point, he/she is also asked to provide the hardness of this data point, which is assumed to have noise. The paper proposes an algorithm similar to explore-then-commit to solve the above problem and theoretically prove the upper bound of the algorithm. It also proposes another meta-algorithm that assumes less knowledge of the sparsity of the bandits. Finally, it compares its algorithm with other baselines algorithms using various datasets.

**Strengths:**

1. This paper has a strong theoritical guarantee for the algorithms it propose.
2. It compares its algorithm with various datasets, ranging from image, texts, and traditional ml datasets.

**Weaknesses:**

1. The motivation and problem setting confuse me, especially for the paragraph from line 67 to line 76. Does the label-scarce regime only applies to the assumption that 'each arm can be pulled at most once'? What are other specialities about this regime. This regime is also kind of broad as many active learning framework is under the assumption that the label is scarce so we want to actively choose valuable data points to sample. Can the authors also elaborate on what the exact use cases of the setting considered in the paper can be applied to the recommendation of perconalized products as described in that paragraph?
2. The assumption that the human will provide noisy hardness is valid, but given this assumption, why do the authors not consider the labels provided by the human expert are also noisy? Can the authors provide more insights or explicit explanations on possible noisy labels?
3. It also feels that the labels provided by the human expert is irrelevant in the problem setting as both the problem formulation and algorithm 1 focuses on getting the human feedback for the hardness $r$ rather than mentioning about the labels. If that is the case, will it be possible to just asking the users for the hardness of the datapoint? How will this affect the algorithm?

**Questions:**

1. What does a reliable trained model mean, does it mean the training data is 100% accurate or something else?
2. why this reliable trained model is absent in the label-scarce setting?
3. what kind of label does the human expert provide to the model? binary or multi-class?
4. Perhaps it is trivial, can the authors explain why the noise $\eta_t$ disappears from equation 1, is it due to condition 1 in line 191? But since equation 1 is not an expectation term, it confuses me.
5. can the authors explain the technical difficulty in the lower bound, though it mentions that it is an open problem in the end. What is the "most likely" lower bound for this problem? As the authors mention that the upper bound could be improved to $T^{\frac{1}{2}}$.

---

> ### Author Response · Authors · 2024-11-16
> **Response to Review (Part 1)**
>
> We'd like to thank the reviewer for appreciating our strong/novel theoretical guarantees and for the constructive feedback. Please note our detailed clarifications to the questions raised below:
>
> **The motivation and problem setting .. Can the authors also elaborate on what the exact use cases of the setting considered in the paper can be applied to the recommendation of personalized products as described in that paragraph?**
>
> We apologize for the confusion in L67-76. Due to space restrictions, we could not expand on that paragraph. We feel that the best clarification is to point to some other published theoretical works that have motivated the blocking constraint ('each arm can be pulled at most once') while being motivated purely based on personalization in recommendation - (see Bresler et al. 2014 and Pal et al. 2024 in paper)
>
> Let us elaborate below. Consider a movie recommendation system and a particular user for which the goal is to personalize. The user will typically watch a movie once, or even if they watch a movie multiple times, it is unlikely that their rating for the movie will change. If we think of movies as arms with unknown mean rewards, then the user feedback on a particular arm will provide only a noisy sample from the reward distribution. The goal is to learn the user preferences from the observed ratings provided by the user - however, we will not get multiple i.i.d. ratings for the same movie. This is where our framework can be applicable - the user is modeled by a sparse linear function with unknown parameters, and the movies by arms. We demonstrate this application on several datasets in Appendix A.2.1.
>
>
> We agree that active learning has the same objective, but it suffers from a cold start period -  active learning techniques need a large pool of samples with gold labels to begin with (see Li et al. 2024 in paper) - otherwise, the confidence
> signals itself are way too noisy, we therefore provide an alternate approach for annotating in a label-scarce regime where an expert annotator is available and can only annotate each sample once.
>
> The focus of the paper is on the label-sparse regime with an expert annotator, where the number of total annotations available is very limited, and there is a single expert annotator available for each data sample. The key specialty of the regime, where each arm can be pulled once, is in the practicality of the scenario where the learning task is complicated, and ground truth labels are needed from an expert annotator due to confidentiality or expertise reasons. Such a scenario restricts the number of times a sample can be annotated.
>
> **The assumption that the human will provide noisy hardness is valid, but given this assumption, why do the authors not consider the labels provided by the human expert are also noisy  Can the authors provide more insights or explicit explanations on possible noisy labels?**
>
> Excellent question! Note that for any niche downstream task, some data with gold labels has to be collected via human experts having domain knowledge - all we are saying is that the hardness scores for annotation are collected from these experts. We agree that the datapoints with gold labels itself can be noisy but that is a challenging problem in itself and is outside the scope of this paper.
> crowd-sourcing platforms. To avoid confusion, we do consider that the hardness scores provided by experts are  noisy - please see L101-103 where $\eta$ denotes the i.i.d noise.
>
> **It also feels that the labels provided by the human expert is irrelevant in the problem setting as both the problem formulation and Algorithm 1 focuses on getting the human feedback for the hardness r rather than mentioning about the labels. If that is the case, will it be possible to just asking the users for the hardness of the datapoint? How will this affect the algorithm?**
>
> This is a good observation and requires some discussion. Note that, for a downstream task, the end goal is to collect task-specific labels (say writing a gold summary for input document for summarization task) via a human expert annotator who is providing gold labels - the main question is which data points to query for labels. Intuitively, from the perspective of the expert annotator, providing the hardness feedback is a small overhead to providing the actual task-specific label (summary). Alternatively, time taken can also be a proxy for the hardness feedback. The hardness feedback is used to select more difficult and informative samples for annotation. Just asking for the hardness would not get the labels (which is the primary goal of annotation) since experts' time is expensive - however, we agree that only asking for the hardness scores does not affect the algorithm.

---

> ### Author Response · Authors · 2024-11-16
> **Response to Review (Part 2)**
>
> **Questions**
>
> **What does a reliable trained model mean, does it mean the training data is 100\% accurate or something else?**
>
> Apologies for the confusion. With the phrase "reliably train model", we mean a model that has been trained on a sufficient number of data points to have good generalization properties and confidence intervals. A model that is trained on too few datapoints will be overfitted and be noisy/unreliable.
>
> **Why is this reliable trained model absent in the label-scarce setting?**
>
> Consider downstream tasks with complex/large output domains wherein a large language model has not seen domain knowledge. Training a reliable model for meaningful confidence intervals requires a significant amount of seed data to begin with, which is not available a-priori (expensive to obtain). Finally, as the number of classes (complexity of output space) increases, the data requirement increases for reasonable confidence or generalization capability [1]. This is not possible in the label-scarce setting.
>
> [1] Yi Yang, Zhigang Ma, Feiping Nie, Xiaojun Chang, and Alexander G. Hauptmann. Multi-class active learning by uncertainty sampling with diversity maximization. Int. J. Comput. Vision, 113(2):113–127, June 2015.
>
> **what kind of label does the human expert provide to the model binary or multi-class?**
>
> To clarify, there are two types of labels that we have introduced in the paper - 1) the label to the downstream task datapoints which can be complex (ex: gold summary for legal documents) - these labels are not used in the algorithm 2)
> For the difficulty score feedback on annotating a task-specific datapoint, the hardness score provided by a human expert is modelled to be a numeric value (see L100-L101 in the paper).
>
>
> **Perhaps it is trivial, can the authors explain why the noise $\eta$ disappears from equation 1, is it due to condition 1 in line 191? But since equation 1 is not an expectation term, it confuses me.**
>
> Equation 1 is the standard definition of regret in the bandit literature [2]. One can think of the definition of regret already containing an expectation over the noise term but not the randomness in the algorithm. The additional expectation in L119 is the expectation with respect to the randomness in the algorithm (as mentioned in the paper).
>
> [2] Tor Lattimore and Csaba Szepesv´ari. Bandit Algorithms. Cambridge University Press, 2020.
>
> **Can the authors explain the technical difficulty in the lower bound, though it mentions that it is an open problem in the end  What is the "most likely" lower bound for this problem? As the authors mention that the upper bound could be improved to $T^{1/2}$.**
>
> We have now been able to prove a tight $\Omega(\min(k^{1/3}\mathsf{T}^{2/3},\sqrt{d\mathsf{T}}))$ lower bound on the regret for our problem by a reduction of the unblocked setting (studied in Hao et al.) to the blocked problem and subsequently invoking the lower bound in Hao et al. - this lower bound matches the regret upper bound achieved for hard sparse vectors in the blocked setting (see the result in Theorem 2 in paper with tail $\beta_k=0$).  To see the detailed proof, please take a look at Appendix A.1.1 in the updated paper. We emphasize that in a sparse linear bandits framework, the dependence of $\mathsf{T}^{2/3}$ is actually tight. The $\mathsf{T}^{1/2}$ order stated in the Conclusion, can be achieved but not without more restrictive assumptions such as knowledge of the minimum signal of the parameter vector similar.

---

> > ### Comment · Reviewer_MzN1 · 2024-11-27
> >
> > Thanks to the authors for providing the feedback to my questions. I would suggest adding the explanation of the motivation in the introduction so the readers could understand the problem better. Also I tend to think that since the main focus is to learn the hardness feedback from the users, asking for the data labels sound redundant and it won't affect the main results for this paper. I have bumped up my score and good luck with the submission.

---

> > > ### Author Response · Authors · 2024-11-27
> > > **Thanks!**
> > >
> > > We thank the reviewer for increasing their score.
> > >
> > > We will definitely expand on the motivation as the reviewer suggests.
> > >
> > > We completely agree that from the point of view of the paper and its main theoretical results, asking for the data labels is redundant. All we are saying that in practice, for any complex downstream task (summarization for instance), some data needs to be assigned gold labels (gold summaries) by a domain expert - during this process, asking for hardness scores (for annotation) is a small overhead. We will add this as a remark for clarification in the paper.
> > >
> > > Thanks again!
> > >
> > > Authors

---

> ### Author Response · Authors · 2024-11-22
>
> We'd like to thank the reviewer for their detailed feedback on our paper. The reviewer raised some valid points, and we have addressed them in our rebuttal. Specifically, we have derived a lower bound which shows that our algorithm is order-optimal.
> In the remainder of the discussion period, we would love to answer any remaining questions or concerns that the reviewer and would love to provide any clarifications if required.
> We again thank the reviewer for their time and look forward to a productive discussion.

---

### Official Review · Reviewer_bSGh · 2024-11-03

**Soundness:** 2
**Presentation:** 2
**Contribution:** 2
**Rating:** 3
**Confidence:** 4

**Summary:**

This paper studied a sparse linear bandit problem with an additional blocking constraints, i.e., no arm can be pulled more than once. The authors developed an explore-then-commit-type of algorithm which achieves a T^{2/3} regret guarantee with known sparsity level (and under certain assumptions). The authors also developed a corralling algorithm to deal with cases without knowing the sparsity level.

**Strengths:**

It is nice to see that the authors develop their theoretical guarantees considering the effect of the tail magnitude $\beta_k$ at sparsity level $k$. The authors also provide a corralling algorithm to deal with cases without knowing the sparsity level.

**Weaknesses:**

1. While the authors spend some efforts in trying to formulate their problem as a data labeling problem with a small labeling budget, I felt such setting is different from the problem the authors actually studied --- a sparse linear bandit problem with an additional blocking constraints. For instance, the objective of the proposed algorithm is to label data points that are hard to label to minimize the regret (covering the space was not the objective even though the proposed algorithm did that in order to minimize regret). But the objective of data labeling should be to learn a good classifier/regressor, which is inconsistent with your definition of the regret. Why not just formulating the problem as a sparse linear bandit problem?
2. The proposed algorithm only achieves a T^{2/3}-type of regret guarantee, which could be sub-optimal as a \sqrt{T}-type of guarantee is expected. Or the authors should provide a lower bound indicating that their guarantee is near-optimal in their setting.
3. In experiments, the proposed algorithm is completed in two rounds: exploration and exploitation. What about other active learning algorithms? Additionally, how do the other baselines incorporate the feedback on the hardness level? I'm also curious why the method of labeling all data points is outperformed by you algorithm in the hard-valid case.

**Questions:**

See above.

---

> ### Author Response · Authors · 2024-11-16
> **Response to Review**
>
> We thank the reviewer for acknowledging our technically challenging theoretical contributions and for the constructive feedback. To the best of our understanding, the main reason for the low score is the lack of a lower bound which was slightly tricky - however, we have now been able to prove a tight $\Omega(\min(k^{1/3}\mathsf{T}^{2/3},\sqrt{d\mathsf{T}}))$ lower bound on the regret for our problem by a reduction of the unblocked setting (studied in Hao et al.) to the blocked problem and subsequently invoking the lower bound in Hao et al. Hence, our results are indeed order optimal with the additional blocking constraint. We emphasize that in sparse linear bandits framework, the dependence of $\mathsf{T}^{2/3}$ is actually tight (unlike most other bandit settings where the ideal dependence is $\sqrt{\mathsf{T}}$ as the reviewer rightly pointed out).
>
> **While the authors spend some efforts in trying to formulate their problem .... Why not just formulating the problem as a sparse linear bandit problem?**
>
> We agree that the ultimate goal is to train a model for the downstream task, such as summarizing legal documents, where the output space is large and complex. In such cases, simple classifiers or regressors are insufficient, and large ML models are often required. However, a key challenge is how to collect and label data when labeling is expensive due to the scarcity of expert annotators. For instance, writing summaries for legal documents is itself a time-intensive and costly process.
>
> Our approach focuses on identifying which samples (e.g., legal documents) should be labeled by experts to maximize the utility of a limited labeling budget. Existing literature highlights that labeling "hard" samples is critical for improving model performance (see Maharana et al. and Sorscher et al. in the paper). The hardness of a sample, being a numerical value, is easier to model than the full complexity of the downstream task, which makes the sparse linear bandit framework a natural fit. Minimizing regret in this framework ensures that we identify and label as many truly hard samples as possible within the given budget.
>
> The blocking constraint addresses the practical limitation that an expert can only label a sample once, as revisiting the same sample for the same expert does not make sense. Similar constraints have also been modeled in recommendation systems (see Bresler et al. 2014 and Pal et al. 2024 in paper). While our approach frames the problem as a sparse linear bandit problem with blocking constraints, this framing complements the broader goal of data labeling for downstream tasks. By focusing on hard samples, we aim to create a labeled dataset that is particularly valuable for training models capable of handling complex output spaces.
>
>
> **2. The proposed algorithm only achieves a $T^{2/3}$-type of regret guarantee, which could be sub-optimal as a $\sqrt{T}$-type of guarantee is expected. Or the authors should provide a lower bound indicating that their guarantee is near-optimal in their setting.**
>
> As we have mentioned, we have now been able to prove a tight lower bound of $\Omega(\min(k^{1/3}\mathsf{T}^{2/3},\sqrt{d\mathsf{T}}))$ on the regret guarantee in the blocked setting - this lower bound matches the regret upper bound achieved for hard sparse vectors in the blocked setting (see the result in Theorem 2 in paper with tail $\beta_k=0$. To see the detailed proof, please take a look at Appendix A.1.1 in the updated paper.
>
>
> **In experiments, the proposed algorithm is completed in two rounds: exploration and exploitation. What about other active learning algorithms? Additionally, how do the other baselines incorporate the feedback on the hardness level? I'm also curious why the method of labeling all data points is outperformed by your algorithm in the hard-valid case.**
>
> To the best of our knowledge, we are the first to propose an algorithm that tries to identify hard samples with the help of experts who are providing gold labels themselves. That is, we are unaware of any other baseline that incorporates the feedback on hardness level. Moreover, in experiments, we have compared with two state-of-the-art active learning baselines, namely  (AnchorAL and SEALS). We emphasize that the main contributions of our work are theoretical, and experiments provide a sound validation of the theory.
>
> Regarding the performance of the model trained on all data points, we agree that it is an interesting observation from the reviewer. Our intuition is that the skewed nature of the dataset leads to this phenomenon.
> Most of the points in the dataset are easy data points (90\%) that are spread uniformly in the vector space - this leads to the trained model being biased towards the easy data points - very similar to standard disease detection datasets where the class imbalance leads to poor performance on the positive class validation dataset.

---

> ### Author Response · Authors · 2024-11-22
>
> We appreciate the feedback provided by the reviewer and thank the reviewer for acknowledging our theoretical novelty. We have answered the questions and comments raised by the reviewer in our rebuttal. Primarily, we have provided a lower bound which shows that our algorithm is order-optimal in our setting.  We would love to use the discussion period to address any other concerns that the reviewer has and provide any clarification if required.
> We again thank the reviewer for their time and effort to help improve our submission.

---

> > ### Comment · Reviewer_bSGh · 2024-11-28
> >
> > I'd like to thank authors for their responses.
> >
> > However, I still find the connection between the sparse linear bandit formulation and the data labeling problem to be weak, especially the bandit formulation mainly focuses on labeling hard data points instead of learning a good classifier. Consider a situation where there are many nearly redundant hard data points. Your bandit formulation will try to label all of these hard data points despite their similar feature representations. This approach may fail to provide the diversity and coverage needed for learning a good classifier, particularly in the data scarce setting you studied.
> >
> > I'd like to keep my rating and recommend a major revision of the current submission.

---

> ### Author Response · Authors · 2024-11-28
> **Response**
>
> Good point!
>
> Can we request the reviewer to look at insight 2 in the paper? Our algorithm handles diversity and coverage already. Specifically, we'd like to point out that the first phase of our algorithm specifically samples from a set which has a near optimal maximum minimum eigenvalue ($\lambda_{\min}$ of the subset is nearly as large as could be) [Theorem 3 in Paper]. Sampling from this subset is shown to result in a diverse enough covering the set [Theorem 6 in paper]. Therefore, even if there were multiple similar hard samples only a few would be picked in this subset and hence, our algorithm does account for diverse datapoint selection.
> Additionally, one straightforward way to handle this during exploitation as well is to cluster datapoints during exploitations and only consider the cluster centers for selecting the hard cluster to be annotated. Then simply annotate one of the datapoint from the selected cluster. We can add a remark regarding the same and it is an interesting future work direction to look at more sophisticated mathematical formulation for the constraint.

---

> > ### Comment · Reviewer_bSGh · 2024-11-28
> >
> > My point is that the connection between the sparse linear bandit formulation and the data labeling problem is weak: the bandit formulation tries to label hard data points to minimize regret, which is not necessarily aligned with the goal of learning a good classifier. How you design your algorithm to solve the bandit problem is a separate issue. I suggest a revision on the either the bandit formulation or the data labeling problem you try to study.

---

> > > ### Author Response · Authors · 2024-11-28
> > >
> > > We thank the reviewer for being responsive and engaging with us.
> > > >"My point is that the connection between the sparse linear bandit formulation and the data labeling problem is weak"
> > >
> > > The sparse linear bandit framework theoretically and practically ensures an efficient data labeling procedure. We have tried our best to make the connection exact between the two. The title, abstract and problem formulation (Section 1.1) and our claims are consistent with this. Our problem formulation gives a one-to-one map between the two. We'd love to discuss if there is something unclear or unrealistic in the formulation and would love to improve the same.
> > >
> > > >"the bandit formulation tries to label hard data points to minimize regret, which is not necessarily aligned with the goal of learning a good classifier. "
> > >
> > > In the paper we motivate labeling hard data points for learning a good classifier (Introduction) and give a brief literature review (Related Work) of established areas of machine learning including curriculum learning and coreset selection which rely on the same hypothesis.  We'd also like to mention an SVM analogy where a very good classifier can be learnt by _only_ considering points close to the decision boundary, which are also the _most difficult_ datapoints.
> > >
> > > We'd like to clearly state that our proposed bandit formulation is a novel approach that handles practical constraints when annotating data points in an industrial or niche setting where only a single expert annotator is available and only a few annotations can be done.
> > >
> > > Thanks again for engaging with us and looking forward to a further productive discussion.

---

> > > ### Author Response · Authors · 2024-11-29
> > > **Alternate Explanation**
> > >
> > > Let us try to clarify in a different way. The reviewer's main concern is that connection between the sparse linear bandit formulation and the data labeling problem is weak: the bandit formulation tries to label hard data points to minimize regret, which is not necessarily aligned with the goal of learning a good classifier.
> > >
> > > **Please note that annotating **hard** datapoints in the data-scarce regime has been established empirically as one of the critical desiderata of collecting data (see Maharana et al. and Sorscher et al. in the paper). This has been the motivation behind our framework wherein we try to collect as many hard datapoints as possible for building a good "classifier" in the end.** Of course diversity and good coverage is also important - all we are saying that algorithmically minimizing the regret naturally handles diversity too (Insight 2 in paper).
> > >
> > > **We also request the reviewer to carefully note the detailed experiments on real datasets where we showcase the efficacy of our algorithms and framework. Here we highlight the limitations of active learning baselines.**
> > >
> > > **Final note** We have also motivated this work for recommendation too (see response to Reviewer MzN1).

---

### Official Review · Reviewer_dhzP · 2024-11-05

**Soundness:** 3
**Presentation:** 2
**Contribution:** 2
**Rating:** 5
**Confidence:** 3

**Summary:**

The paper addresses the challenge of efficiently annotating data points under the constraints of limited annotation rounds in a label-scarce environment. It proposes a novel methodology that integrates expert feedback on the difficulty of annotating specific data points, leveraging a sparse linear bandits framework. This approach focuses on selecting the most informative samples to annotate, which optimizes the use of scarce expert resources by prioritizing data points that are both challenging and representative. Theoretical results show the sub-linear regret of the proposed BSLB algorithm.

**Strengths:**

1. The application of sparse linear bandits to annotation in a label-scarce environment addresses a significant practical problem in machine learning, particularly in situations where acquiring labeled data is expensive or logistically difficult.
2. Introducing blocking constraints into the bandit problem formulation is novel and aligns well with practical scenarios where data points cannot be repeatedly annotated.
3. This paper provides a rigorous theoretical analysis on the regret which quantifies the efficiency of the BSLB algorithm. This analysis is backed by proofs that demonstrate how the algorithm effectively balances exploration and exploitation under sparsity and blocking constraints.

**Weaknesses:**

1. It would be beneficial to make a more thorough comparison with the works that do not assume blocking constraints. Is there any instance where the blocking constraints would clearly fail for those existing algorithms like Hao et al. (2020)?
2. It would be good to move the definition and description of regret being concerned earlier in the paper. It might confuse the readers with the discussion on the regret without knowing what regret is being considered.

**Questions:**

Is it possible to make some modifications to the existing sparse linear bandit algorithm to accommodate the blocking constraints? What are the key difficulties that the blocking constraints add to the problem?

---

> ### Author Response · Authors · 2024-11-16
> **Response to Review**
>
> We thank the reviewer for recognizing the paper's key contributions: the novel application of sparse linear bandits to label-scarce annotation problems with practical blocking constraints, alongside the rigorous theoretical analysis of BSLB algorithm's regret that demonstrates effective exploration-exploitation balance.
> Please see our clarifications below for the concern raised:
>
> **1. It would be good to move the definition and description of regret being concerned earlier in the paper. It might confuse the readers with the discussion on regret without knowing what regret is being considered.**
>
> Apologies for the confusion. We have now moved the definition and description of regret along with the preliminaries earlier in the updated paper, as the reviewer suggested before our main contributions.
>
> **2  It would be beneficial to make a more thorough comparison with the works that do not assume blocking constraints. Is there any instance where the blocking constraints would clearly fail for those existing algorithms like Hao et al. (2020)?**
>
> This is a great suggestion - it is easy to modify the existing algorithm ESTC proposed in Hao et al. by incorporating the blocking constraint. In ESTC, the authors have first computed a distribution over the arms, sampling from which will provide good coverage of the arms - they do so in Step 4 for some rounds (Explore), and then in Step 8, they greedily choose the best action repeatedly for remaining rounds (Exploit). Note the simple modification to the exploration component - instead of sampling directly from the computed distribution in Step 4 of ESTC, we can employ rejection sampling where we re-sample from the distribution until a unique arm is sampled (respects the blocking constraint). However, this completely changes the distribution according to which the arms are sampled - think of the special case when the computed distribution in ESTC puts the entire probability mass on just a few arms (say $d$ arms).  However, due to the blocking constraint, once those $d$ arms are pulled, we will have to resort to pulling arbitrary arms - therefore, instead of getting approximately $\text{exploration rounds}/d$ noisy responses for each of the $d$ chosen arms, we just get $1$. This completely breaks the ensuing statistical guarantees proved for ESTC regarding the lasso estimator. It is clear that in this special case, Step 4 of ESTC will fail. We hope this demonstrates the challenge involved. We will add this as a remark in the paper if the reviewer suggests it.
>
> In fact, incorporating the blocking constraint for the similar objective of finding a good arm cover implies a reformulation of the optimization problem in ESTC - the reformulation leads to a discrete optimization problem (Eq. 2 on Page 5 of our paper), which is non-convex - a naive brute force search is computationally infeasible (exponential in the number of arms).   One of our main novel contributions is to find a good approximation algorithm to the discrete optimization problem (see Section 2.3 in the paper).
>
> **Is it possible to make some modifications to the existing sparse linear bandit algorithm to accommodate the blocking constraints  What are the key difficulties that the blocking constraints add to the problem?**
>
> We answer the first part of the question in the preceding paragraph. We highlight that apart from the additional blocking constraint, we provide statistical guarantees for the soft sparsity setting when the data points only satisfy the weak assumption of RE (Restricted Eigenvalue). Both the blocking constraint and soft sparsity entail several technical challenges (summarized in the Technical Challenges paragraph (L153-L172) in the paper).

---

> ### Author Response · Authors · 2024-11-22
>
> We'd again like to thank the reviewer for their comments and acknowledging the strong theoretical contributions of the paper.
> We have addressed all the comments and questions raised by the reviewer in our reply.  We have also made the requested changes in the paper.
>
> We'd love to engage with the reviewer in the remainder of the discussion period to address any other comments or provide any clarification.  We want to ensure that the contributions of our work are clear so that the reviewers can make a fair evaluation. We thank the reviewer again for their time and effort in helping us improve our contribution. Looking forward to a productive discussion.

---

> > ### Comment · Reviewer_dhzP · 2024-11-28
> >
> > I thank the authors for the responses. After reading other reviews and responses, I will keep my score.

---

> > > ### Author Response · Authors · 2024-11-28
> > > **Request to elaborate**
> > >
> > > Did we resolve the concerns of the reviewer? If that is the case, we are a bit surprised that the reviewer decided not to increase their score.
> > >
> > > Can the reviewer elaborate more on the concerns in "other reviews/responses" - it would really help our paper and we will appreciate it.
> > >
> > > Thanks
> > >
> > > Authors

---

### Author Response · Authors · 2024-11-18
**Clarifications and Responses provided**

We thank the reviewers for the considerable time and effort that they put into reviewing our work. We are glad that every reviewer has found our theoretical contributions to be strong. We have provided answers/clarifications to all the questions that were raised. in particular, several reviewers asked about a lower bound - we have also been able to prove that the lower bound in the data-scarce regime for blocked sparse linear bandits is $\Omega(\mathsf{T}^{2/3})$ (when parameter vector is hard sparse) just as in the case without blocking constraint (Hao et al.) Hence our algorithmic regret guarantees are order optimal.

We request the reviewers to kindly look at the responses and please let us know if their other questions asked have been answered. We will be happy to provide further clarifications if required.

Thanks again
Authors

---

### Meta-Review · Area_Chair_K2zo · 2024-12-11

**Metareview:**

This paper proposes a novel approach to the data annotation problem under a label-scarce environment using a sparse linear bandit with a blocking constraint. The Explore-Then-Commit (BSLB) algorithm aims to minimize regret by annotating the hardest data points, with theoretical guarantees and empirical validation showing its effectiveness.

Reviewers appreciated the novelty of applying sparse linear bandits to active learning but raised concerns about the weak connection between minimizing regret and the goal of data labeling. The focus on hardness feedback instead of labels was seen as misaligned with typical active learning objectives. The theoretical analysis was viewed as similar to existing work. There were also calls for a clearer problem objective, more comparisons with other active learning methods, and better handling of model misspecification and noisy labels.

**Additional Comments On Reviewer Discussion:**

After discussions, the authors have addressed some of the issues. However, the reviewers still believe that the connection between the sparse linear bandit and data labeling is weak. Additionally, the approach closely resembles existing work, and the problem-setting feels unnatural since hardness feedback alone could suffice.

---

### Decision · Program_Chairs · 2025-01-22

Reject